# TexEditor: Structure-Preserving Text-Driven Texture Editing

**Bo Zhao** [1] [*]  **Yihang Liu** [2]  **Chenfeng Zhang** [3]  **Huan Yang** [4] [†]  **Kun Gai** [4]  **Wei Ji** [1] [†]

## Abstract

Text-guided texture editing aims to modify object appearance while preserving the underlying geometric structure. However, our empirical analysis reveals that even SOTA editing models frequently struggle to maintain structural consistency during texture editing, despite the intended changes being purely appearance-related. Motivated by this observation, we jointly enhance structure preservation from both data and training perspectives, and build TexEditor, a dedicated texture editing model based on Qwen-Image-Edit-2509. Firstly, we construct TexBlender, a high-quality SFT dataset generated with Blender, which provides strong structural priors for a cold start. Secondly, we introduce StructureNFT, a RL-based approach that integrates structure-preserving losses to transfer the structural priors learned during SFT to real-world scenes. Moreover, due to the limited realism and evaluation coverage of existing benchmarks, we introduce TexBench, a general-purpose real-world benchmark for text-guided texture editing. Extensive experiments on existing Blender-based texture benchmarks and our TexBench show that TexEditor consistently outperforms strong baselines such as Nano Banana Pro. In addition, we assess TexEditor on the general-purpose benchmark ImgEdit to validate its generalization. Our code and data are available at https://github.com/KlingAIResearch/TexEditor.

## 1. Introduction

Replacing object textures or modifying texture attributes such as roughness is a core capability underlying many downstream vision applications, including content genera-

* Work done during an internship at Kuaishou Technology. [1]Nanjing University [2]Shan Dong University [3]Zhejiang University [4]Kolors Team, Kuaishou Technology. Correspondence to: Huan Yang <hyang@fastmail.com>, Wei Ji <weiji@nju.edu.cn>.

*Proceedings of the 43rd International Conference on Machine Learning*, Seoul, South Korea. PMLR 306, 2026. Copyright 2026 by the author(s).

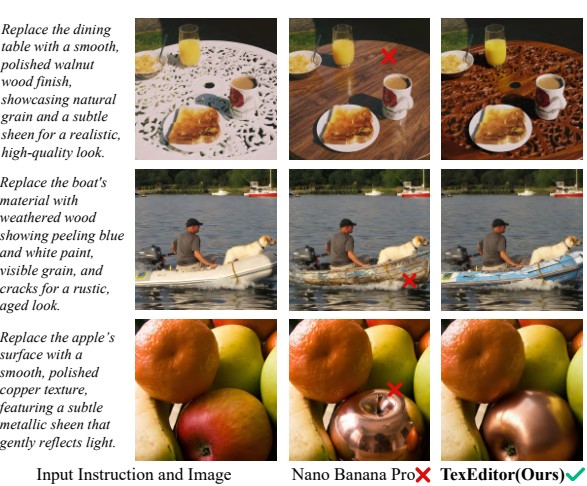

*Figure 1.* Existing image editing models often struggle to preserve structural consistency with the original image during texture editing, and instead tend to regenerate the edited subject.

tion, and virtual scene construction (Huang et al., 2025b; Yeh et al., 2024). Compared to traditional manual editing or rule-based approaches (Khan et al., 2006), text-driven texture editing has attracted increasing attention due to its intuitive and user-friendly interaction paradigm (Guerrero-Viu et al., 2024). However, texture editing remains challenging due to limited paired data (Zhao et al., 2024), subtle yet semantically meaningful texture-induced visual changes, and the requirement to preserve object geometry.

In recent years, image editing techniques have advanced rapidly, driven by powerful generative backbones such as diffusion models (e.g., Stable Diffusion (Rombach et al., 2022)), as well as large-scale training (Wu et al., 2025). Existing methods can produce visually plausible texture edits in simple or controlled settings (Cheng et al., 2024), particularly when scenes contain salient foreground objects. However, even commercial systems that achieve state-of-the-art performance across many tasks, such as Nano Banana Pro (DeepMind, 2025b), often fail to preserve fine-grained geometric structure in realistic scene-level texture editing scenarios, as shown in Figure 1.

Motivated by these observations, we introduce TexEditor, which addresses a key gap in existing texture editing sys-

tems: the lack of effective mechanisms to preserve object structure when following appearance-only editing instructions. To this end, we construct TexBlender, a scene-level simulated training dataset built using Blender (Blender Online Community, 2018) and 3D-Front (Fu et al., 2020). Blender provides a controllable rendering environment that allows us to generate paired images where only the target texture is modified while the underlying 3D structure remains unchanged, enabling clean supervision for structure preservation. Crucially, unlike prior texture editing datasets that focus on isolated objects or simple scenes, we incorporate the 3D-Front dataset to introduce diverse and complex scene layouts and backgrounds(Wang et al., 2025), encouraging the model to learn structure-preserving behavior beyond salient foreground objects. Furthermore, during texture editing in Blender, we apply texture changes to grouped sets of 3D objects rather than individual instances, producing training data with varying editing granularity and enabling the model to handle both local and scene-level texture edits. These two design choices set TexBlender apart from existing datasets and provide stronger supervision for learning structure-aware texture editing.

Using this data, we perform supervised fine-tuning (SFT) to teach the model both texture editing skills and the ability to preserve geometric structure. To enable these capabilities to generalize to real-world scenes, we further apply reinforcement learning on texture editing queries constructed from COCO (Lin et al., 2014). Building on existing MLLM-guided reinforcement learning techniques (Sun et al., 2025), we introduce a structure-preserving auxiliary loss (StructureNFT) (Liufu et al., 2025), which leverages low-level structural cues extracted from SAM masks (Kirillov et al., 2023) to strongly enforce invariance of geometric structure during the editing process.

On the evaluation side, we address two key limitations of existing benchmarks. First, most current benchmarks are based on Blender-generated single-object synthetic scenes (Sharma et al., 2024), which are visually unrealistic and fail to capture practical editing challenges such as occlusion, small objects, and complex backgrounds. To overcome this, TexBench, a new benchmark built on COCO, is constructed. We first leverage instance-level mask annotations to precisely localize editing targets, then employ Qwen3-VL-32B (Bai et al., 2025) to automatically generate diverse and semantically coherent texture editing instructions, followed by human verification to ensure consistency in instruction validity and editing difficulty. Secondly, existing evaluation metrics often rely on MLLM-based scoring of high-level instruction adherence (Chen et al., 2025), which tends to overlook unintended structural changes even when prompts explicitly emphasize that such changes are undesirable. To address this, we propose TexEval, a composite metric that combines low-level visual structural cues with MLLM instruction adherence, providing a more holistic assessment of texture editing quality. User studies demonstrate that TexEval reliably reflects both semantic correctness and geometric structure preservation, outperforming existing automatic metrics.

Extensive experiments are conducted to evaluate TexEditor. Beyond standard comparisons on Blender-based texture benchmarks and our TexBench, TexEditor consistently outperforms strong baselines such as Nano Banana Pro. We also perform ablation studies to examine the impact of cold-start SFT, the quality of cold-start data, structural loss, and its regularization. The results demonstrate that each design choice contributes significantly to the model's final performance, validating the rationale behind our approach. Furthermore, on texture-related sub-tasks of the general-purpose ImgEdit benchmark (Ye et al., 2025b), TexEditor, fine-tuned from Qwen-Image-Edit-2509 (Qwen-2509) (Wu et al., 2025), surpasses its base model, indicating strong potential to generalize beyond texture-specific tasks while preserving geometric structure.

Our core contributions can be summaried as follows:

- We introduce TexBench and TexEval, providing a more realistic and comprehensive evaluation dataset, along with an evaluation metric that jointly accounts for instruction following and structure preservation.

- We design a two-stage training strategy that enhances structure preservation during the editing process without modifying the architecture of existing image editing backbones, and propose the TexBlender dataset together with the StructureNFT training method to support this objective.

- Our TexEditor consistently outperforms existing strong baselines such as Nano Banana Pro on both Blender benchmarks and TexBench, while preserving structural consistency and generalizing well to texture-related sub-tasks in ImgEdit.

## 2. TexEditor

We study the task of text-guided texture editing, where the goal is to modify the appearance-related texture attributes of a target object in an image according to a natural language prompt, while keeping the object's geometry, spatial layout, and semantic identity unchanged and minimizing unintended changes to non-target regions. Formally, given an input image $I$ and a prompt $P$, the model produces an edited image $I_e$. It can be formulated as follows:

$$I_e = Model(I, P). \tag{1}$$

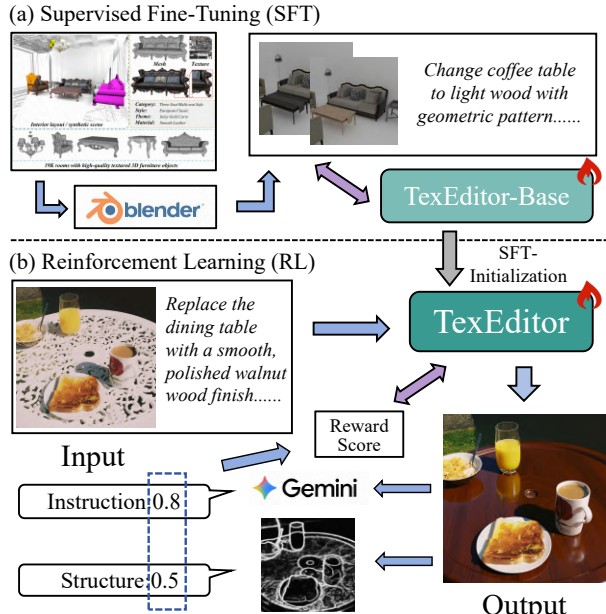

(a) Supervised Fine-Tuning (SFT)

(b) Reinforcement Learning (RL)

*Figure 2.* The training pipeline of **TexEditor**. (a) The model undergoes Supervised Fine-Tuning (SFT) on a synthetic dataset rendering 3D assets in Blender. (b) It is then optimized via Reinforcement Learning (RL), utilizing a multi-faceted reward system that incorporates Gemini for instruction alignment and structural analysis to balance texture editing with structural preservation.

To address the limitations of existing text-driven texture editing models in preserving geometric consistency, we propose a structure-aware approach that integrates both data construction and training strategies. We first generate high-quality, scene-level texture editing supervision data using Blender (Blender Online Community, 2018), where pre- and post-edit images share identical scene structure and object geometry, with changes confined to target texture attributes, explicitly decoupling texture from structural variations. To enhance real-world generalization, we further employ a reinforcement learning stage on real-scene interactions with a hybrid objective: an instruction adherence reward to ensure accurate texture edits, and a geometry consistency loss derived from edge detectors to enforce structural invariance, effectively suppressing the model's tendency to re-generate objects while achieving high-fidelity, structure-preserving edits, which is illustrated in Figure 2.

## 2.1. TexBlender dataset

Motivated by our empirical observation that existing image editing models often fail to preserve object and scene structure during texture editing, we identify a lack of effective training signals for structure-preserving appearance manipulation. To address this gap, we construct TexBlender, a high-quality scene-level simulated dataset for texture editing. Leveraging Blender's fully controllable rendering pipeline, TexBlender provides paired images in which only the target

textures are modified while the underlying 3D geometry remains unchanged, enabling clean supervision for structural preservation.

Crucially, unlike prior texture editing datasets that focus on isolated objects or visually simple scenes, TexBlender integrates the 3D-Front dataset to introduce diverse and complex indoor layouts with rich backgrounds, encouraging models to learn structure-preserving behavior beyond salient foreground objects. Furthermore, texture edits are applied to grouped sets of 3D objects rather than individual instances, producing training data with varying editing granularity and enabling both local and scene-level texture edits. Together, these design choices distinguish TexBlender from existing datasets and provide stronger supervision for structure-aware texture editing. A high-level description of the method is presented below, while full implementation details can be found in Section B.

**Scene Preparation and Editing Target Definition** We first curate scenes from 3D-FRONT to ensure geometric and physical plausibility, removing objects with severe interpenetration, degenerate geometry, or inconsistent scale. To support texture editing at multiple semantic granularities, we define editing targets through grouping strategies based on object semantics and texture identifiers, aggregating furniture sub-components into coherent units. For each selected target group, we fix the camera viewpoint and render a reference image, ensuring that all paired samples share identical scene geometry and differ only in appearance.

**Texture Variation Rendering** Texture edits are performed on the target furniture using the Principled BSDF shader (Burley & Studios, 2012). We consider two complementary editing modes: (1) attribute-level adjustments, where roughness, metallicity, and transparency are modified according to the object's overall texture state; and (2) global texture replacement, where high-quality textures from MatSynth (Vecchio & Deschaintre, 2024) are applied to generate distinct appearances. Each edit is rendered independently, and original texture parameters are restored after rendering to avoid cross-sample interference.

**Instruction Generation and Semantic Alignment** During rendering, we record detailed texture modification metadata, including edit type, adjustment direction, and applied textures. Based on these metadata, Qwen3-VL (Bai et al., 2025) is used to generate diverse natural language texture editing instructions. To ensure precise alignment between language and visual content (Wei et al., 2025), we further refine instructions using vision-guided grounding with SAM3 (Carion et al., 2025), correcting references and mitigating hallucinated descriptions (Huang et al., 2025a; Li et al., 2025a).

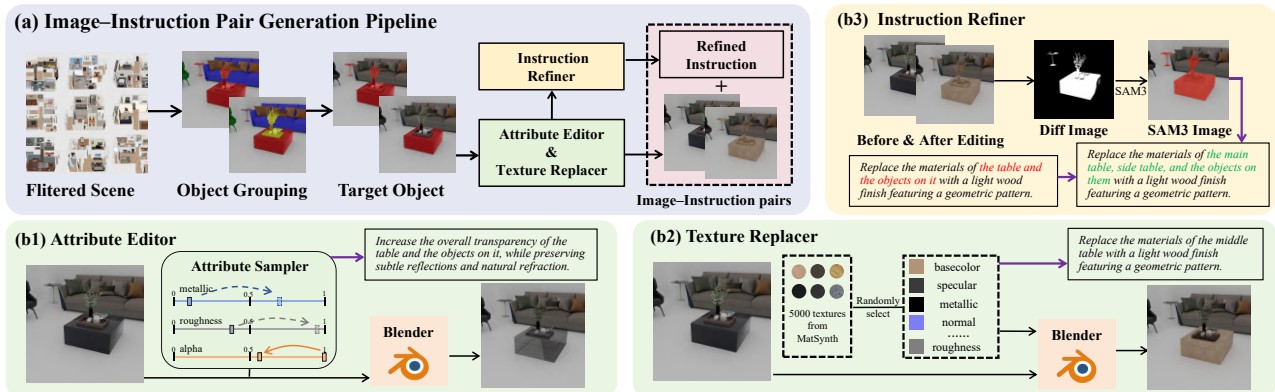

*Figure 3.* Generation of image-instruction pairs in **TexBlender**. We first identify target object groups within filtered 3D scenes. Visual variations are created via (b1) Attribute Editing (adjusting shader parameters) or (b2) Texture Replacement (applying MatSynth textures). Finally, (b3) an Instruction Refiner leverages visual cues (difference images) and segmentation masks (SAM3) to generate precise, grounded text instructions that accurately describe the texture changes.

## 2.2. Structure-aware RL on Real Images

While the simulated SFT data TexBlender provides strong and explicit supervision, its visual realism and distributional diversity are inherently limited. Given that our SFT backbone already demonstrates strong base capabilities, we therefore adopt reinforcement learning to generalize the structural preservation learned during SFT to real-world scenes.

**Training Data** The RL stage is conducted on training data constructed using the same methodology as TexBench, as described in Section 3. This dataset comprises two complementary sub-tasks: *attribute-level texture modification* (e.g., adjusting roughness or metallicity) and *texture replacement* (e.g., converting wood to metal). By combining these tasks, the model is exposed to both subtle and substantial appearance changes, promoting robust learning across a range of editing scenarios.

**StructureNFT** Considering the high computational cost of reinforcement learning, the recently proposed, low-cost DiffusionNFT (Zheng et al., 2025) is adopted as the baseline. For the reward signal, texture editing requires both precise adherence to instructions and preservation of the original structure. Therefore, in addition to the instruction-following loss, a structure-preserving loss term is introduced to balance the trade-off between maintaining geometric consistency and following the editing instruction. It can be formulated as:

$$Reward = Score_{ins} + Score_{struct}. \tag{2}$$

Specifically, $Score_{ins}$ is formulated as:

$$Score_{ins} = MLLM(I_e, I, P, P_{sys}), \tag{3}$$

where $P_{sys}$ represent the systrem prompt of instruction score and it can be found in the appendix. Considering

both computational efficiency and accuracy of reward computation, Gemini 3 Flash (DeepMind, 2025a) is employed as the MLLM for instruction-following loss (Zheng et al., 2023). And $Score_{struct.}$ is formulated as:

$$Score_{struct} = Dist(E(I_e), E(I)), \tag{4}$$

Where, $D$ denotes a certain distance metric, and $E$ refers to the extractor of structural information from the image.

For the structural consistency score, we investigate three variants that differ in the choice of the structure extractor $E$ and the distance function $Dist$. (1) Mask-based shape IoU, computed from SAM3 segmentations (Carion et al., 2025), which measures coarse object-level shape consistency; (2) Wireframe-based IoU, which evaluates the overlap of predicted and reference structural layouts; and (3) Wireframe-based SSIM (Wang et al., 2004), which assesses structural similarity on wireframe representations at a finer granularity. As shown in the first row of Figure 4, the SAM3-based extractor captures object structure at a coarse semantic level, while wireframe-based extractors such as SAUGE (Liufu et al., 2025) provide more detailed geometric representations. However, as illustrated in the second row of Figure 4, wireframe-based IoU is highly sensitive to minor pixel-level perturbations introduced during the editing process, leading to large score fluctuations even when the underlying structure remains largely unchanged. In contrast, SSIM is more robust to small spatial misalignments, making it a more reliable distance metric for assessing structural consistency in practice.

While SSIM provides fine-grained and perturbation-robust measurements of structural consistency, its raw scores $s$ tend to be compressed within a narrow range, as texture editing typically affects only localized regions of the object structure. To improve metric sensitivity, we introduce task-specific empirical normalization schemes tailored for

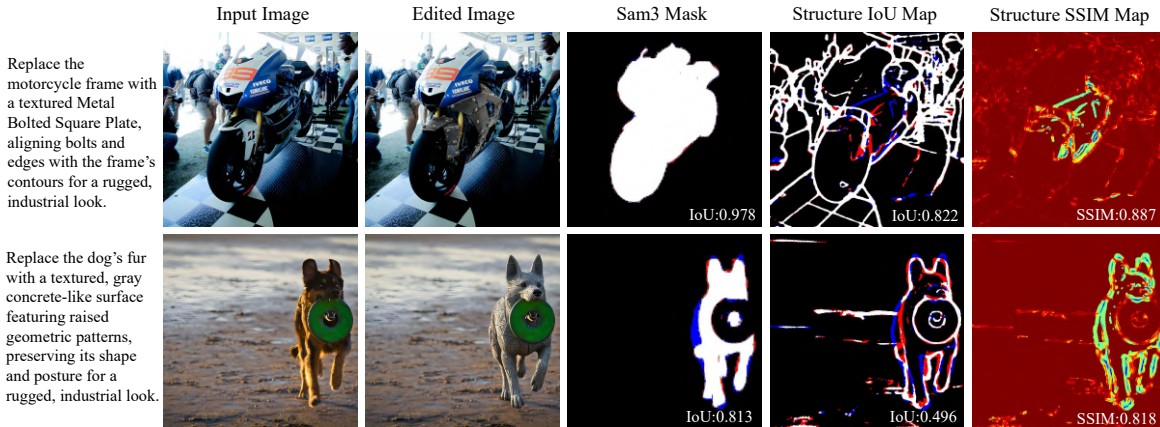

*Figure 4.* Visualization of different structure extractors and distance metrics in structure consistency loss

texture replacement and attribute editing, respectively, and validate their effectiveness through ablation studies. The corresponding formulations are as follows:

$$s = SSIM(SAUGE(I_e), SAUGE(I)), \quad (5)$$

$$\phi(s; \tau_{min}, \tau_{max}) = \begin{cases} -0.2, & \text{if } s < \tau_{min} \\ 1, & \text{if } s > \tau_{max} \\ \dfrac{s - \tau_{min}}{\tau_{max} - \tau_{min}}, & \text{otherwise} \end{cases} \quad (6)$$

$$Score_{struct}^{subtask} = \phi(s; \tau_{min}^{subtask}, \tau_{max}^{subtask}), \quad (7)$$

where, $\phi$ represents the normalizing function, and $subtask$ denotes one of the two texture editing sub-tasks. Further details are provided in Section A.1. Notably, the threshold scores used in the normalization are determined based on empirical observations. While their effectiveness is validated through ablation studies, optimality is not guaranteed.

# 3. TexBench: A Real-World Benchmark for texture Editing

During the development of our method, we observe significant limitations in existing evaluation datasets and protocols (Sharma et al., 2024). As summarized in Table 1, prior benchmarks exhibit clear deficiencies along several critical dimensions, including task coverage, data realism, and dataset scale, which makes them insufficient for a comprehensive assessment of current texture editing methods. To address these issues, we construct TexBench, a high-quality benchmark for real-world scene texture editing.

For open-ended tasks such as texture editing, LLM-based evaluation has become the dominant paradigm (Basu et al., 2023). However, as illustrated in Figure 5, such evaluators are inherently constrained by their own perceptual and rea-

soning capabilities: even advanced models Gemini-3 (Deep-Mind, 2025a) can not reliably capture subtle, unintended structural changes introduced during texture editing. This observation is consistent with recent studies showing that the visual reasoning ability of MLLMs remains imperfect, especially when reliable assessment requires fine-grained visual grounding, temporal dynamics, or explicit reasoning over visual changes(Li et al., 2026; Liu et al., 2026b). To address this limitation, we introduce TexEval, an evaluation protocol that jointly considers instruction fulfillment and structural consistency by explicitly incorporating structural information. In addition, we conduct a user study to validate the effectiveness and reliability of the proposed evaluation.

## 3.1. TexBench Construction

When constructing TexBench, we explicitly address the limitations of existing evaluation datasets in terms of realism, scene complexity, and scale, and design the benchmark accordingly (Basu et al., 2023). Specifically, we construct the benchmark on COCO (Lin et al., 2014) to ensure diverse, real-world scenes. Instance masks are used to precisely locate editable objects, and the object's label is provided to Qwen3-VL (Bai et al., 2025) to generate texture editing instructions. This automation pipeline reduces hallucinations compared to generating instructions from scratch while maintaining instruction diversity and sample sufficiency. More details can be found in Section D.

## 3.2. Evaluation metrics

For texture editing, a highly multi-modal and underdetermined image generation task, existing evaluation typically relies on large multimodal models for scoring (Sharma et al., 2024). In preliminary experiments, it was observed that even SOTA models such as Gemini-3 fail to reliably detect and reflect subtle structural changes between the original and edited images, even when prompts explicitly emphasize

*Table 1.* Comparison with existing texture editing benchmarks.

| Name | Protocol | Task | Data Source | Scene | Size | Quality Control |
|---|---|---|---|---|---|---|
| Alchemist | Not released | Attribute modification | Blender (synthetic) | Single-object | 150 | Not specified |
| TexBench | Open-source | Attribute + replacement | COCO (real images) | Real-world | 825 | Manual inspection |

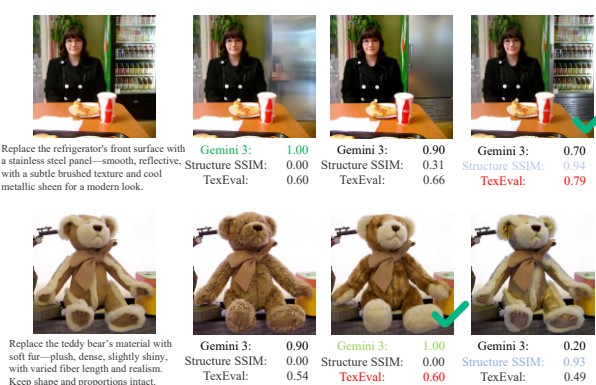

*Figure 5.* **Qualitative Comparison of Different Evaluation Metrics.** Results favored by Gemini, structural, and TexEval scores are shown in green, blue, and red; check marks indicate human preference. TexEval balances the structure and semantics and aligns best with humans.

structural consistency, as shown in Figure 5. To address this limitation, low-level structural visual information is incorporated to complement model-based evaluation. Specifically, structural signals derived from the RL training process are combined with large-model scores to form the proposed evaluation metric, TexEval, which can be formulated as:

$$TexEval = \alpha * Score_{Ins} + (1 - \alpha) * Score_{struct.}, \quad (8)$$

where $\alpha$ represents the balancing hyperparameter. Although this linear combination is simple, our user study shows that it outperforms both standalone instruction-following and purely structure-based metrics, aligning more closely with human preferences (Figure 5). A preference study with five annotators over 500 pairwise comparisons is conducted to quantitatively analyze metric performance and determine the final weights. Further Detail are provided in Section E.

## 4. Experiment

We conduct a series of experiments to evaluate our proposed TexEditor. We organize our experimental analysis into four parts: (1) experimental setup, including datasets, baselines, and evaluation metrics; (2) performance comparison with existing methods on both real-world and synthetic blender datasets; and (3) ablation studies to analyze the contributions of different components, including scene-level Blender data, supervised fine-tuning (SFT), and reinforcement learning (RL) with structural constraint; (4) generalization study of TexEditor on the general-purpose editing

dataset ImgEdit (Ye et al., 2025b).

### 4.1. Experimental Setup

**Baselines** We categorize the compared methods into two groups: (1) large foundation models, including proprietary and open-source general-purpose editing systems (Deep-Mind, 2025b; Wu et al., 2025); (2) existing texture-editing methods, such as Alchemist (Sharma et al., 2024).

**Datasets** In our experiments, we use TexBlender for cold-start SFT training, containing 5,000 instruction-edit pairs evenly split between texture replacement and attribute editing subtasks. For reinforcement learning, we employ 1,500 COCO-based edited queries, comprising 500 for texture edits and 1,000 for attribute edits. Evaluation is conducted on TexBench, which consists of high-quality, human-verified texture editing queries, including 453 texture replacement examples and over 372 attribute editing examples.

**Evaluation Metrics** We employ three metrics: instruction following scoring, structural consistency score and the proposed TexEval metric. TexEval is designed to jointly assess adherence to textual instructions and preservation of geometric structure.

**Implementation details** We develop our model based on the Qwen-2509 (Wu et al., 2025) using the Edit R1 codebase (Li et al., 2025b). During inference, all commercial models use their default settings, while our model employs the DPM++ solver (Lu et al., 2022) with 20 steps. Further detail are provided in Section A.1.

### 4.2. Comparative Evaluation

Comparative experiments were conducted on both the proposed TexBench and the existing Blender-based benchmark. In the TexBench setting, we reproduce the previous academic work, Alchemist, and further built upon it using the latest editing foundation, Qwen-2509, comparing it against our TexEditor as well as several commercial open- and closed-source models. The results are shown in Figure 6. Most of these baselines exhibit varying degrees of poor instruction adherence and structural degradation of the edited objects. Nano Banana Pro (DeepMind, 2025b) partially mitigates these issues; however, it may still alter local object characteristics, such as the content displayed on the computer in the first row or the Microsoft logo on the laptop

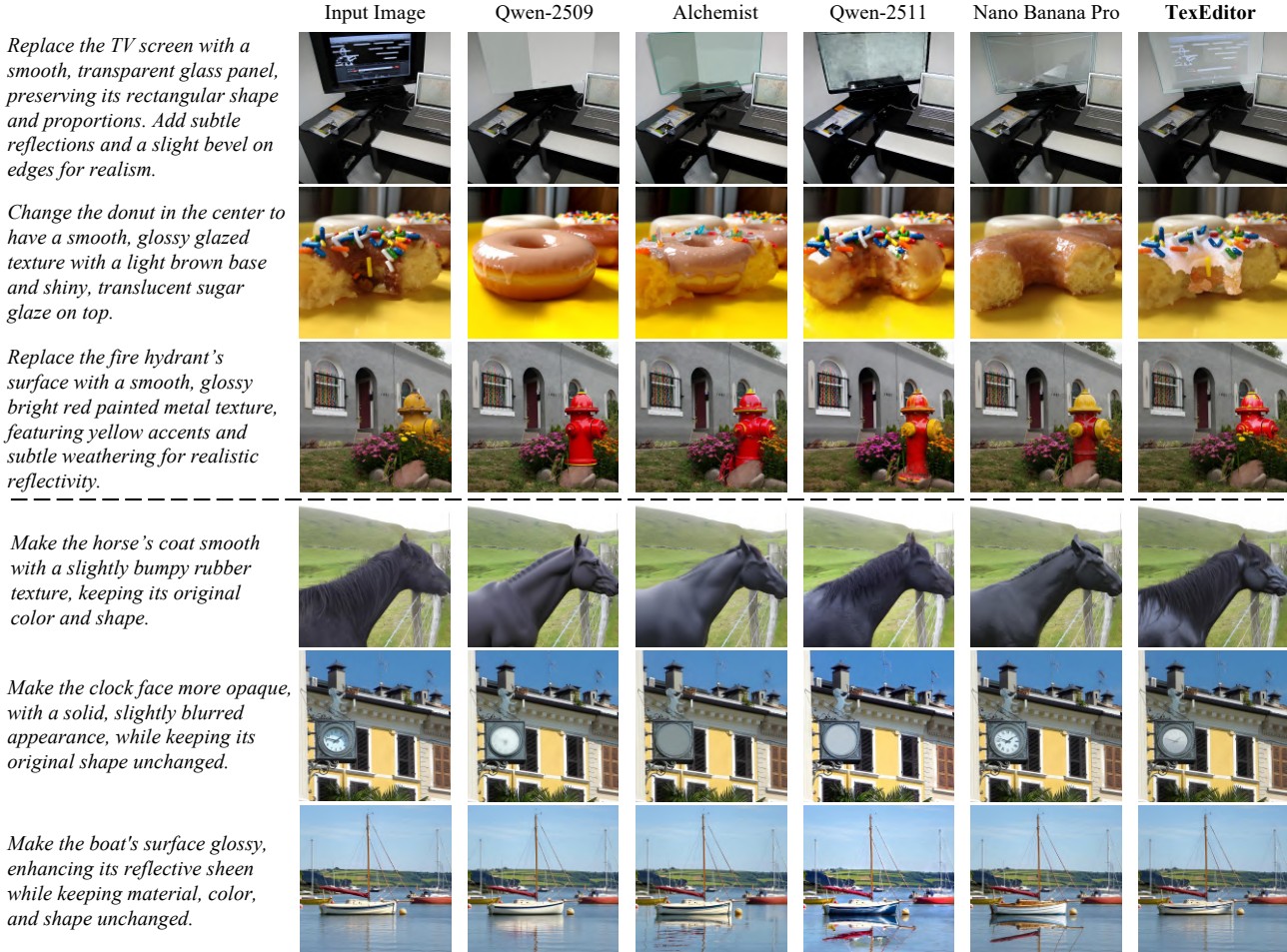

| | Input Image | Qwen-2509 | Alchemist | Qwen-2511 | Nano Banana Pro | **TexEditor** |

*Replace the TV screen with a smooth, transparent glass panel, preserving its rectangular shape and proportions. Add subtle reflections and a slight bevel on edges for realism.*

*Change the donut in the center to have a smooth, glossy glazed texture with a light brown base and shiny, translucent sugar glaze on top.*

*Replace the fire hydrant's surface with a smooth, glossy bright red painted metal texture, featuring yellow accents and subtle weathering for realistic reflectivity.*

*Make the horse's coat smooth with a slightly bumpy rubber texture, keeping its original color and shape.*

*Make the clock face more opaque, with a solid, slightly blurred appearance, while keeping its original shape unchanged.*

*Make the boat's surface glossy, enhancing its reflective sheen while keeping material, color, and shape unchanged.*

*Figure 6.* **Qualitative Comparison.** TexEditor achieves the best visual results by modifying texture according to instructions while preserving the target object's structure.

in the fifth row. In contrast, our method demonstrates high instruction adherence while effectively preserving structural consistency, which is also validated by the quantitative results presented in Table 2.

In addition to our proposed TexBench, we reproduce the single-object Blender-based benchmark introduced in Alchemist, and further extend it with a texture replacement sub-task. The results are reported in Table 3. Due to limited computational resources, we compare against only a few key baselines, including the training backbone Qwen-2509 and Nano Banana. Our method consistently achieves superior performance in both instruction-following accuracy and structure preservation. Because the dataset is rendered with environment maps and 3D assets, the resulting images contain relatively few structural edges, leading to higher variance in structure-related metrics. More qualitative results are provided in Section A.2.

*Table 2.* Quantitative comparison cross texture and attribute editing tasks within the TexBench

| Method | Texture | | | Attribute | | |
|---|---|---|---|---|---|---|
| | Inst. ↑ | Structure ↑ | TexEval ↑ | Inst. ↑ | Structure ↑ | TexEval ↑ |
| Qwen-2509 | 0.767 | 0.642 | 0.717 | 0.584 | 0.408 | 0.514 |
| ALchemist | 0.761 | 0.711 | 0.741 | 0.631 | 0.512 | 0.583 |
| TexEditor-Base | 0.796 | 0.733 | 0.767 | 0.670 | 0.544 | 0.620 |
| Qwen-2511 | 0.789 | 0.569 | 0.701 | 0.719 | 0.362 | 0.576 |
| Nano Banana | 0.726 | 0.584 | 0.669 | 0.530 | 0.332 | 0.451 |
| Nano Banana Pro | 0.839 | 0.801 | 0.824 | 0.716 | 0.697 | 0.708 |
| TexEditor | **0.858** | **0.929** | **0.886** | **0.723** | **0.816** | **0.760** |

### 4.3. Ablation Studies

Our ablation study investigates the roles of simulated SFT data and reinforcement learning with different loss components, with results summarized in Table 4. We consider three data configurations: training without SFT data, with the single-scene single-object simulated data proposed by Alchemist, and with our multi-granularity simulated data covering complex scenes. Comparing Configs A–C under

*Table 3.* Performance comparison cross texture and attribute editing tasks within the Blender-based benchmark

| Method | Texture | | | Attribute | | |
|---|---|---|---|---|---|---|
| | Inst. ↑ | Edge ↑ | TexEval ↑ | Inst. ↑ | Edge ↑ | TexEval ↑ |
| Qwen-2509 | 0.791 | 0.841 | 0.811 | 0.777 | 0.856 | 0.809 |
| Nano Banana | 0.855 | 0.497 | 0.712 | 0.782 | 0.588 | 0.704 |
| Nano Banana Pro | 0.923 | 0.752 | 0.855 | 0.865 | 0.883 | 0.872 |
| TexEditor | **0.961** | **0.988** | **0.972** | **0.965** | **0.982** | **0.972** |

no reinforcement learning, Alchemist data provides moderate improvements over the base model, but its simplified scenes limit the diversity of texture editing patterns that can be learned, resulting in weaker gains in instruction adherence and structural preservation compared to our multi-granularity data. Further analysis of cold-start initialization on RL (Configs D, E, and H) shows that stronger cold-start model consistently leads to better RL outcomes.

On the RL side, we analyze the effects of different RL loss designs, including instruction-following loss derived from Gemini 3, structure-preservation loss based on computer vision tools, and a regularization strategy applied to the structure loss. Experiments with Configs F and G, where only instruction-following loss or structure-preservation loss is applied, yield moderate improvements but introduce clear trade-offs. Combining instruction-following loss with an unregularized structure-preservation loss (Config H) fails to fully exploit structural supervision. In contrast, our full model (TexEditor, Config I) integrates instruction-following loss with a properly regularized structure-preservation loss, achieving strong and balanced performance in both instruction adherence and structural consistency.

### 4.4. Hyperparameter Sensitivity and Data Quantity

To further evaluate the sensitivity of the reward design, structural-loss hyperparameters, and RL data quantity, we conduct a set of controlled experiments, as shown in Table 5. Unless otherwise specified, all settings use the same RL training budget: 40 update steps, 24 prompts per step, and 6 rollouts per prompt. We denote the default configuration as Base-40.

The first analysis focuses on reward-weight configurations and reward APIs. In Base-40, the instruction and structure rewards are combined with a ratio of 0.55:0.45. We additionally evaluate two shifted reward-weight configurations, denoted as Ins:Struct=3:7 and Ins:Struct=8:2, which place relatively more emphasis on structure preservation and instruction following, respectively. Although these variants change individual metric scores, their effect on the overall TexEval score is limited. We also replace the reward API used in Base-40 with Gemini-2.5-Flash, denoted as Gemini-2.5. This variant leads to only a slight performance drop, suggesting that the method is reasonably robust to the choice

of reward API.

The second analysis examines the sensitivity of the structural-loss thresholds. Since the SSIM-based structure score follows a non-uniform data distribution, with around 80% of Qwen-Edit outputs falling within 0.65–0.90, aligning the threshold design with this range through a smoother thresholding strategy is important for stable optimization. In Base-40, the lower and upper thresholds are set to 0.8 and 0.95 for attribute editing, and 0.7 and 0.9 for texture editing. We further test a relaxed-threshold variant, denoted as Thresh-40, by decreasing the lower threshold by 0.1 for both tasks while keeping the upper threshold unchanged. This relaxation weakens the structural constraint and allows the model to emphasize instruction following more, yet the overall performance remains largely unchanged, indicating robustness to this hyperparameter.

Finally, we investigate whether more RL data and longer training can compensate for removing the structural loss. To this end, we train the model without the structural loss while continuously using new data, so that the model does not complete a full epoch. The variants No-Struct-40, No-Struct-60, and No-Struct-80 denote training without the structural loss for 40, 60, and 80 update steps, respectively. The results show that simply increasing data and computation, even doubling both, still cannot match Base-40, especially on attribute editing tasks where structural preservation is more critical. The improvement also saturates as training progresses. These findings indicate that the benefit of the structural loss cannot be replaced by data scaling alone, and that it is essential for achieving strong and stable editing performance.

### 4.5. Generalization study

TexEditor was evaluated on the general image editing benchmark ImgEdit (Ye et al., 2025b) against Qwen-2509 to test the generality of its structure-preserving and instruction-following abilities. The results show improvements on tasks related to texture editing, such as Background, Adjust, and Compose, indicating effective generalization to out-of-domain datasets. Performance is slightly lower on tasks less related to texture editing, likely due to limited training data coverage rather than model design. Detailed quantitative and qualitative results can be found in Section A.3.

## 5. Related Work

Recent advances in large-scale generative model training (Zhao et al., 2025a; Ye et al., 2025a; Liu et al., 2026a) have substantially improved text-guided image editing, enabling more precise and semantically aligned edits. These advances have further expanded image editing toward fine-grained and challenging sub-tasks, such as texture editing.

*Table 4.* Ablation study on different combinations of SFT data and RL loss.

| Config | SFT Data | | | RL Loss | | | | Texture | | | Attribute | | |
|---|---|---|---|---|---|---|---|---|---|---|---|---|---|
| | None | Simple | Ours | None | Instruction | Structure | Norm. Structure | Instruction↑ | Structure↑ | TexEval↑ | Instruction↑ | Structure↑ | TexEval↑ |
| a | ✓ | | | ✓ | | | | 0.767 | 0.642 | 0.717 | 0.584 | 0.408 | 0.514 |
| b | | ✓ | | ✓ | | | | 0.761 | 0.711 | 0.741 | 0.631 | 0.512 | 0.583 |
| c | | | ✓ | ✓ | | | | 0.796 | 0.723 | 0.767 | 0.670 | 0.544 | 0.620 |
| d | ✓ | | | | ✓ | | ✓ | 0.801 | 0.823 | 0.801 | 0.628 | 0.717 | 0.663 |
| e | | ✓ | | | ✓ | | ✓ | 0.822 | 0.845 | 0.831 | 0.683 | 0.771 | 0.718 |
| f | | | ✓ | | ✓ | | | **0.867** | 0.602 | 0.761 | **0.739** | 0.473 | 0.633 |
| g | | | ✓ | | | | ✓ | 0.752 | **0.941** | 0.828 | 0.557 | **0.908** | 0.697 |
| h | | | ✓ | | ✓ | ✓ | | 0.832 | 0.693 | 0.776 | 0.715 | 0.622 | 0.690 |
| i | | | ✓ | | ✓ | | ✓ | 0.858 | 0.929 | **0.886** | 0.723 | 0.816 | **0.760** |

*Table 5.* Controlled analysis of reward design, structural-loss hyperparameters, and RL data quantity. All settings use 40 update steps unless otherwise specified.

| Config | Attribute Editing | | | Texture Editing | | |
|---|---|---|---|---|---|---|
| | Ins. | Struct. | TexEval | Ins. | Struct. | TexEval |
| *Default setting* | | | | | | |
| Base-40 | 0.625 | 0.864 | 0.733 | 0.821 | 0.898 | 0.856 |
| *Reward-weight and reward-API sensitivity* | | | | | | |
| Ins:Struct=3:7 | 0.541 | 0.943 | 0.722 | 0.748 | 0.964 | 0.845 |
| Ins:Struct=8:2 | 0.714 | 0.736 | 0.724 | 0.804 | 0.827 | 0.814 |
| Gemini-2.5 | 0.623 | 0.851 | 0.726 | 0.803 | 0.882 | 0.839 |
| *Threshold and data-scaling analysis* | | | | | | |
| Thresh-40 | 0.690 | 0.795 | 0.737 | 0.825 | 0.868 | 0.844 |
| No-Struct-40 | 0.704 | 0.363 | 0.551 | 0.802 | 0.728 | 0.769 |
| No-Struct-60 | 0.738 | 0.449 | 0.608 | 0.814 | 0.761 | 0.790 |
| No-Struct-80 | 0.741 | 0.492 | 0.629 | 0.820 | 0.782 | 0.803 |

Alchemist (Sharma et al., 2024) leverages large-scale synthetic data generated with Blender (Blender Online Community, 2018) to achieve precise texture editing in controlled single-object scenes. However, its reliance on simplified environments limits its generalization to complex real-world images. PhyS-EdiT (Cai et al., 2025b) introduces intermediate texture representations to better disentangle texture changes from other visual factors, thereby improving controllability.

In parallel, recent commercial models, such as Qwen-Image-Edit (Wu et al., 2025) and Nano Banana (DeepMind, 2025b), have demonstrated non-trivial texture editing capabilities. Nevertheless, both research and commercial methods still struggle to preserve object structure during texture editing. In practice, texture changes are often entangled with unintended geometric distortions, leading to shape deformation, boundary inconsistency, or loss of structural details between the input and edited images. This structural inconsistency remains a key challenge for text-guided texture editing.

Related tasks such as shadow generation have explored geometry-aware guidance for structural coherence (Liu et al., 2024; Zhao et al., 2025b), yet preserving object geometry during material or texture editing remains an open problem.

## 6. Conclusion

In this paper, we focus on the critical challenge of preserving structure in text-driven texture editing and propose TexEditor, a texture editing model to cope this problem. Specifically, we construct high-quality simulated data, TexBlender, to teach models structure-preserving capabilities and introduce a novel RL method, StructureNFT, to generalize this ability to real-world scenarios. Extensive experiments demonstrate that TexEditor consistently outperforms strong baselines, including Nano Banana Pro. To further advance the field, we present TexBench and TexEval, a benchmark and a evaluation metric designed to address limitations in existing datasets and assessment protocols. We hope our methods, resources, and findings will inspire future research in structure-preserving, text-guided texture editing.

## Acknowledge

This work was supported by the CCF-Tencent Rhino-Bird Open Research Fund.

## Impact Statement

This work presents TexEditor and TexBlender, methods and datasets for structure-aware, text-guided texture editing. The primary goal is to advance research in image editing and generative modeling. While our methods operate on synthetic or publicly available images, potential misuses include unrealistic manipulation of visual content. We note, however, that the scope of our study is limited to controlled research settings, and we do not identify immediate ethical or societal risks that require mitigation beyond standard responsible AI practices.

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

This supplementary material is organized into five parts. First, we provide a more detailed description of the experimental part, including implementation details, additional quantitative and qualitative comparison results, and generalization experiments on the ImgEdit benchmark. Second, we describe the construction process of TexBlender, detailing data generation procedures and design choices. Third, we present a failure case analysis to highlight the limitations of the proposed method. Fourth, we introduce the construction of TexBench and the reinforcement learning training data used in our experiments. Finally, we elaborate on the user study for metric evaluation, including the annotation protocol and analysis details.

## A. Experimental Details and Additional Results

### A.1. Implement detail

For supervised fine-tuning (SFT) of the Qwen-Image-Edit model (Wu et al., 2025), we use a LoRA-based setup on the DIT backbone (Huang et al., 2024). The LoRA rank is set to 64, and gradient checkpointing is enabled to reduce memory usage. Training is performed with a batch size of 6, learning rate 1e-4, and for 3 epochs, using 8 data loader workers.

Our RL-based texture training uses 18 inference steps per sample, a guidance scale of 4.0, and 6 images per prompt for policy learning. Batch size per GPU is set to 6. KL regularization is set to 0.0001, the learning rate is 1e-4, and gradient clipping is applied with a maximum norm of 1.0. LoRA is optionally used for efficient fine-tuning, and exponential moving average (EMA) is enabled to stabilize model updates. For attribute task, the $\tau_{min}$ and $\tau_{min}$ are 0.8 and 0.95. For texture replacement task, the $\tau_{min}$ and $\tau_{min}$ are 0.7 and 0.9.

### A.2. More qualitative results

Due to the limited space in the main paper, we provide additional qualitative comparison results on TexBench as well as on the Blender-based benchmark here.

As for the TexBench, the results of texture replacement and attribute editing are presented in Figure 7 and Figure 8, respectively. It can be observed that the proposed method consistently achieves superior performance in both structure preservation and instruction adherence.

As for the Blender-based benchmark, the results can be found in Figure 9. As can be seen from the figures, although the image content is relatively simple, our method still demonstrates a clear advantage in maintaining structural consistency during the editing process.

For the texture replacement task, as an additional comparison, we further include ZeST (Cheng et al., 2024), a representative image-to-image based texture transfer method, beyond the previously evaluated Nano Banana, Qwen-Image-Edit, and TexEditor series models.

To make ZeST applicable to instruction-based settings, we convert the textual instruction into a texture reference by first extracting the target texture description and then synthesizing a corresponding texture image using Z-Image (Cai et al., 2025a). This synthesized texture image, together with the original input image, is provided to ZeST for texture transfer.

### A.3. Experiment on ImgEdit

To investigate whether the structure-preserving and instruction-following capabilities learned by TexEditor are limited to texture editing tasks, we further evaluated it on the general image editing benchmark ImgEdit (Ye et al., 2025b), comparing against the baseline method Qwen-2509. The results in Table 6 show that for task types more closely related to texture editing—such as preserving object structure while changing the background (Background), adjusting object attributes (Adjust), and handling composite instructions (Compose)—TexEditor achieves significant improvements in human preference alignment. As illustrated in Figure 10, this indicates that the structure-preserving ability learned by the model can generalize effectively to out-of-domain datasets.

At the same time, we observe that TexEditor exhibits slightly lower performance on tasks that differ more substantially from the texture editing paradigm. We attribute this primarily to the absence of mixed general image editing instructions in the training data; this limitation is due to data coverage rather than a flaw in the model architecture, and is therefore not the main focus of this work.

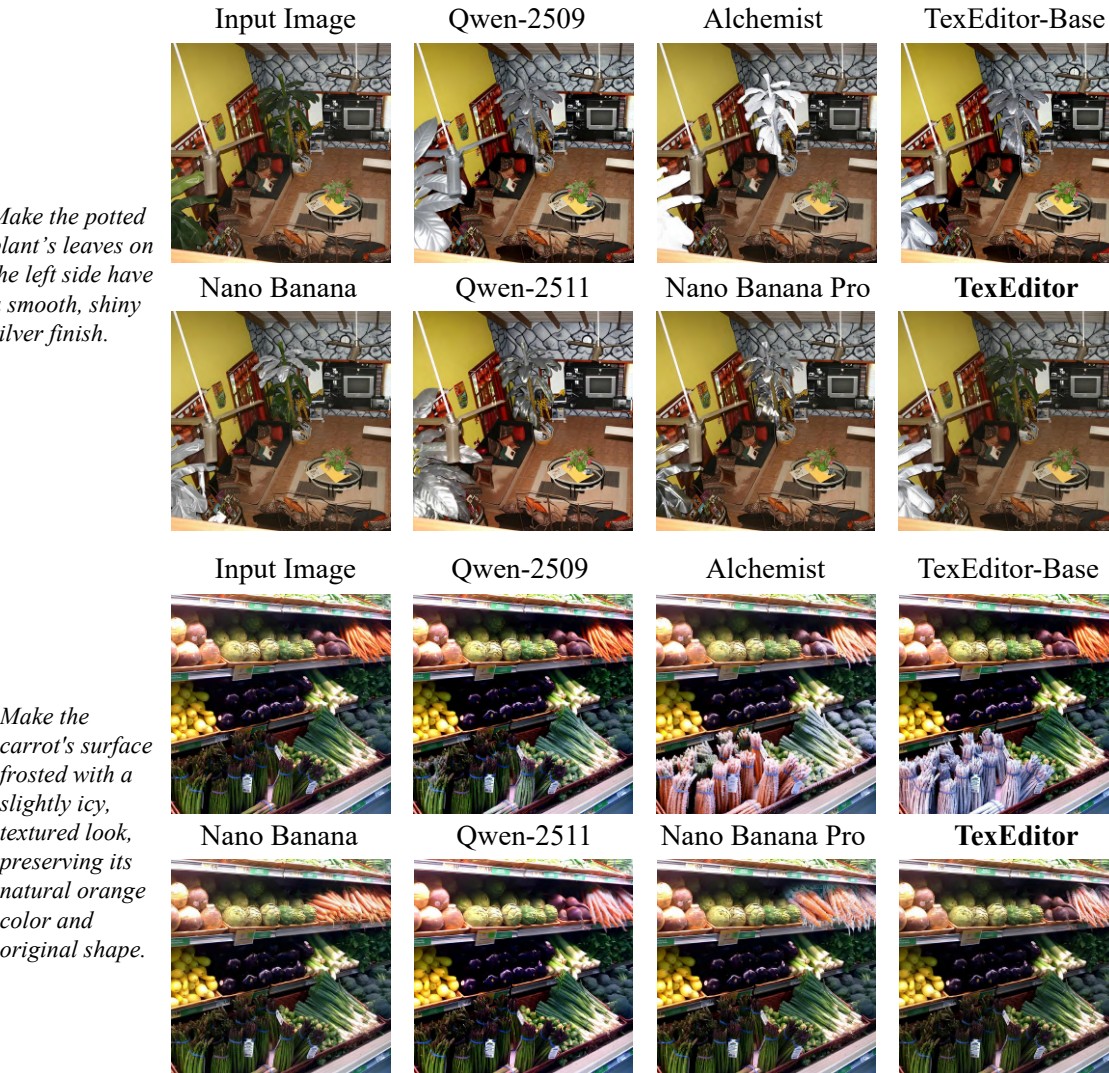

*Figure 7.* Qualitative comparisons on the attribute editing subtask of TexBench

## B. Texture Editing Data Generation Workflow Based on 3D-FRONT and BlenderProc

We adopt the 3D-FRONT dataset (Fu et al., 2020) and the BlenderProc framework (Denninger et al., 2020) to construct a full data generation pipeline for texture editing. The pipeline includes four sequential stages: scene preprocessing, texture variation rendering, vision-guided instruction generation, and quality filtering. Each stage is designed to ensure consistency between visual changes and editing instructions while maintaining high rendering quality.

### B.1. Scene Preprocessing

First, scenes are loaded from the 3D-FRONT room dataset. Although 3D-FRONT provides high-quality indoor layouts, the raw scenes may contain geometric inconsistencies, such as object interpenetration. To address this issue, we perform AABB (Axis-Aligned Bounding Box)–based collision detection to identify spatial conflicts between objects. For each detected conflict, one of the overlapping objects is randomly removed. After preprocessing, the average furniture retention rate is approximately 0.78, which preserves sufficient scene complexity. To further increase scene diversity, the lighting intensity is randomly adjusted during scene loading.

Next, object grouping is performed. In 3D-FRONT, many furniture instances are composed of multiple sub-components (e.g., tabletops and legs) (Fu et al., 2021). We apply two grouping strategies: grouping by object names, and grouping by

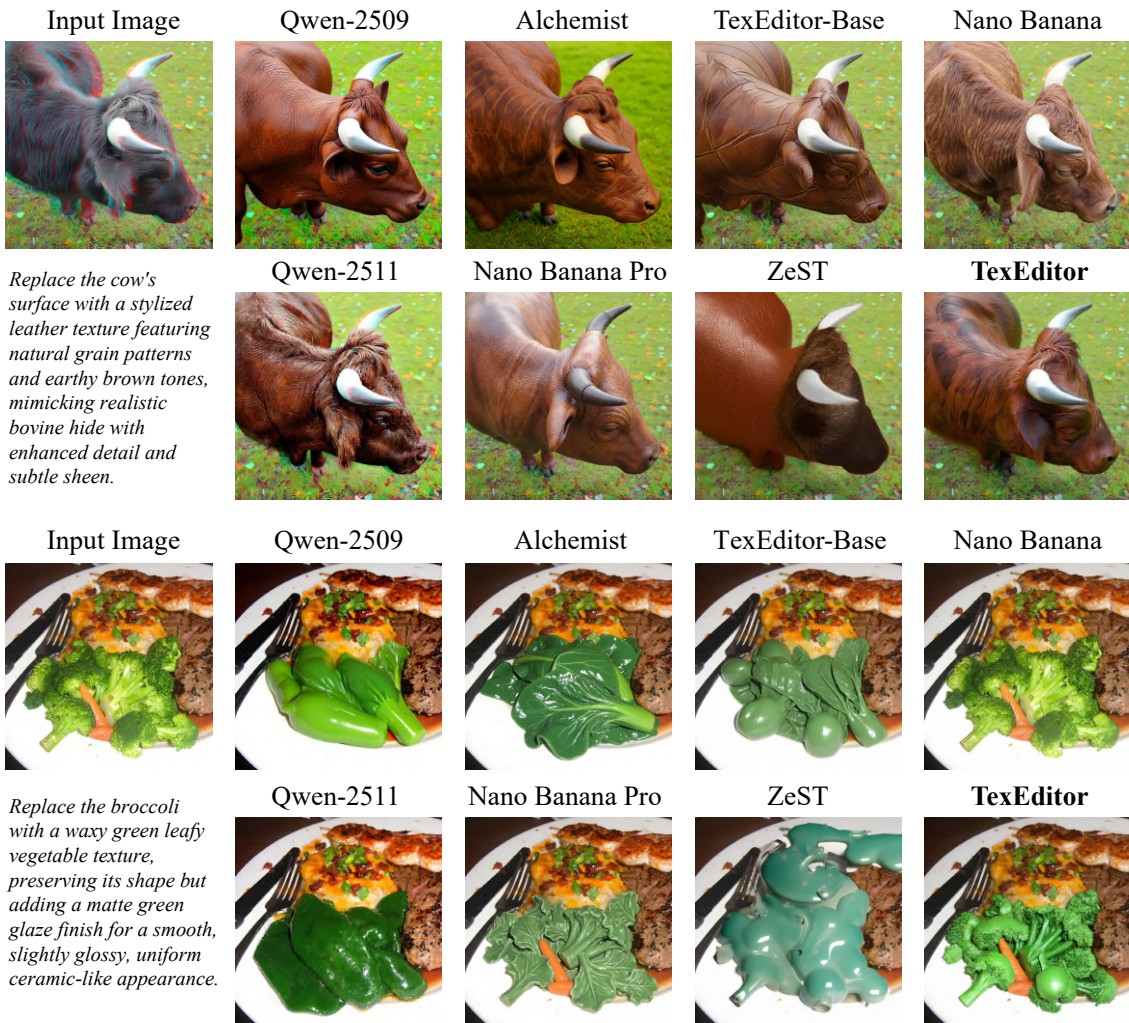

*Figure 8.* Qualitative comparisons on the texture replacement subtask of TexBench. We additionally include ZeST, an image-to-image based texture transfer method, for comparison alongside instruction-based editing models.

the combination of object names and original texture names, where both are provided by 3D-FRONT. This design enables texture editing at different granularities while preserving visual consistency across grouped components, and provides diverse supervision.

After grouping, a target furniture group is randomly selected, and camera poses are sampled around it. Specifically, we compute the world-space bounding box of the target furniture to obtain its center position and overall spatial extent. Camera positions are then sampled within a spherical shell centered at the target, with the radius set to two to four times the maximum dimension of the furniture, and the elevation angle constrained to the range of $[0°, 30°]$. For each sampled camera position, we compute the viewing direction toward the target and construct the corresponding camera transformation matrix. The visibility of the target furniture is evaluated using visibility ratio and ray-based occlusion detection. Up to 100 camera sampling attempts are performed to find a valid viewpoint. Once a viewpoint with sufficient visibility is obtained, the camera pose is fixed and a reference image is rendered at a resolution of $1024 \times 1024$. If no valid viewpoint is found after 100 attempts, the target furniture group is re-sampled and the process is repeated. If this procedure fails for 5 consecutive target selections, the entire room is replaced and the process restarts.

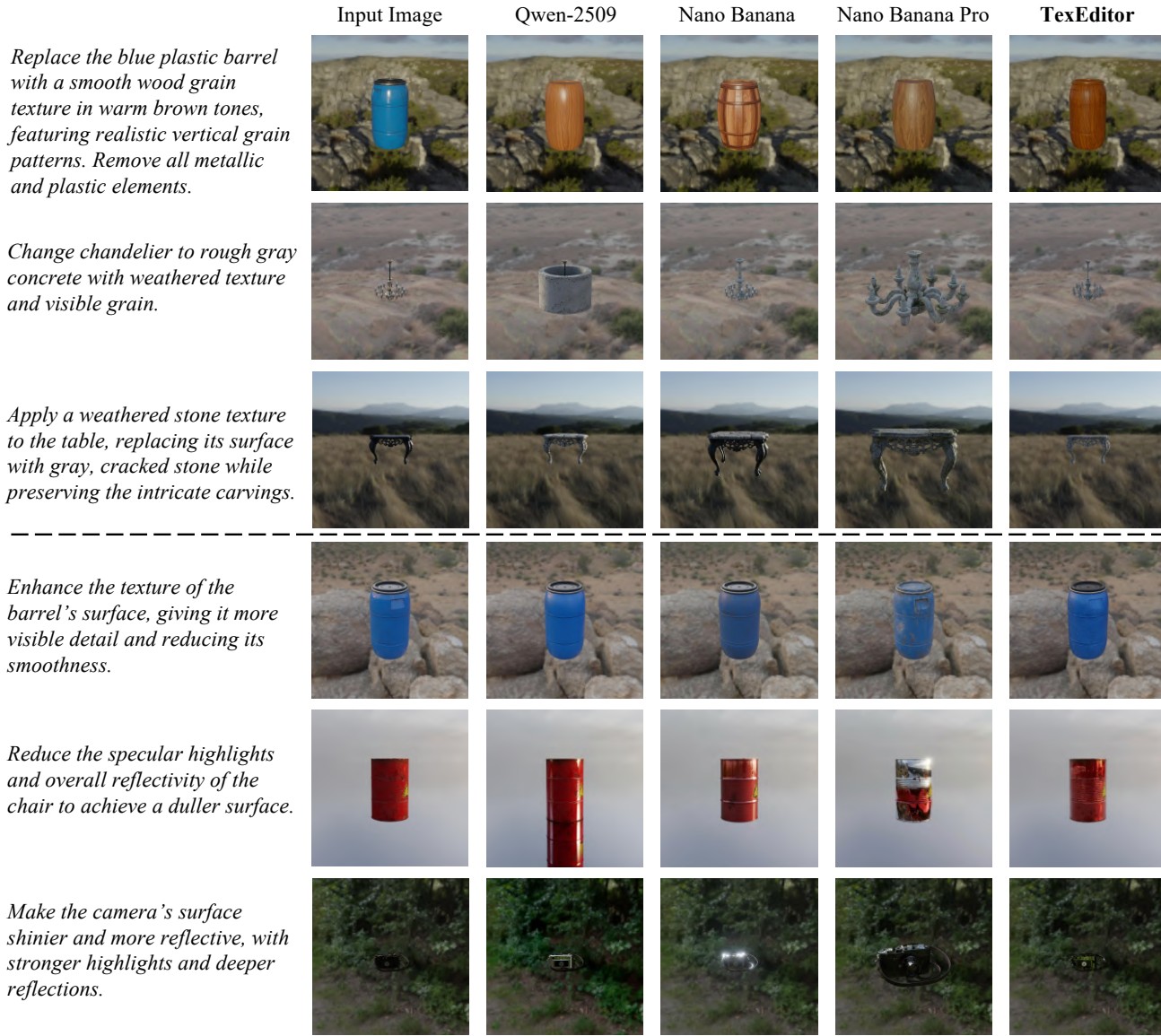

*Figure 9.* Qualitative comparisons on Blender-based benchmarks.

## B.2. Texture Variation Rendering

We apply two types of texture manipulations to the target furniture using the Principled BSDF shader (Burley & Studios, 2012): attribute-level texture adjustment and global texture replacement.

### B.2.1. ATTRIBUTE-LEVEL TEXTURE ADJUSTMENT

For attribute-level texture adjustment, we modify three material properties: Roughness, Metallic, and Alpha. For each property $p$, we traverse all sub-components within the target furniture group $\mathcal{G}$, retrieve the default value from the corresponding input of the Principled BSDF shader node, and compute the average value $\bar{p}$. The adjustment direction is determined based on $\bar{p}$: if $\bar{p} < 0.3$, the property is increased; if $\bar{p} > 0.7$, it is decreased; otherwise, the direction is randomly selected. The adjustment magnitude is fixed to $\pm 0.5$, and the resulting value is clamped to the range $[0.01, 0.99]$. After applying the modification consistently to all components in the group, we render the edited image $I_p$ and then restore the original material parameters. During this process, we record modification metadata, including the object name, original value, updated value, and adjustment direction.

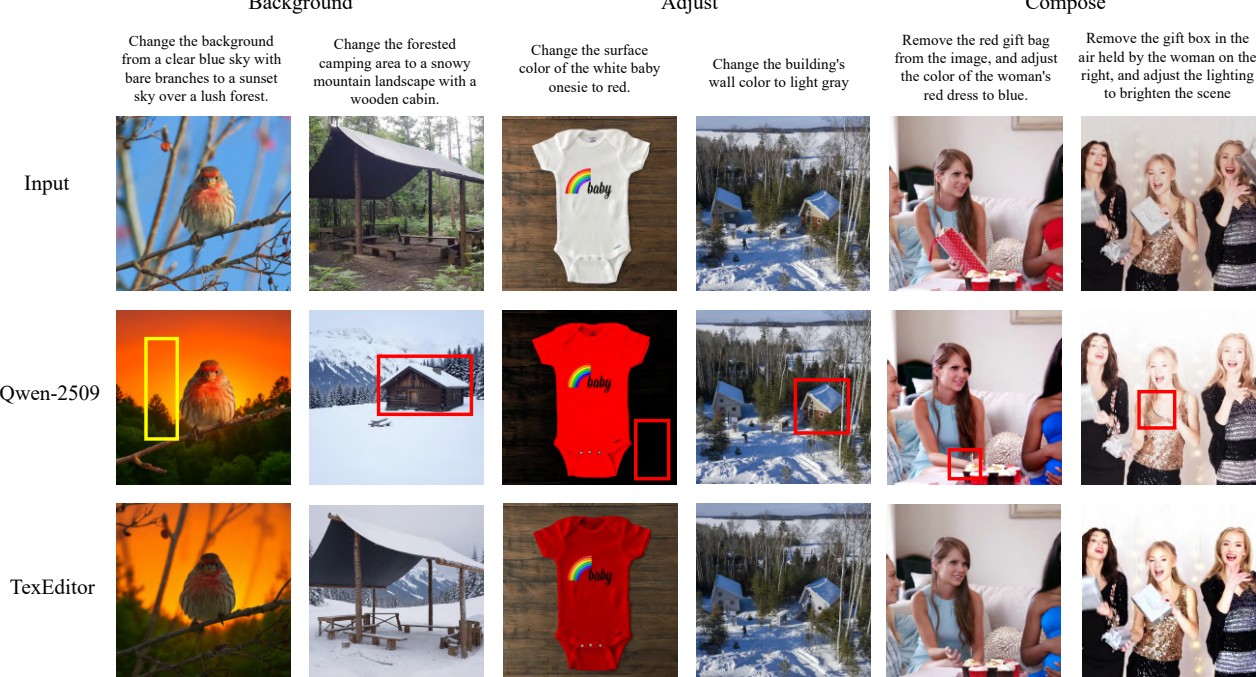

*Figure 10.* Qualitative comparisons on texture-related subtasks of the ImgEdit benchmark

*Table 6.* Scores across different edit types in the ImgEdit benchmark.

| Method | Adjust ↑ | Background ↑ | Compose ↑ | Action ↑ | Add ↑ |
|---|---|---|---|---|---|
| Qwen-2509 | 4.05 | 3.87 | 2.17 | **3.33** | **4.47** |
| TexEditor (Ours) | **4.13** | **4.13** | **2.39** | 2.79 | 4.28 |

### B.2.2. GLOBAL TEXTURE REPLACEMENT

For global texture replacement, we use high-quality PBR textures from the MatSynth dataset (Vecchio & Deschaintre, 2024). In implementation, we identify and load multiple PBR map types, including base color, roughness, metallic, normal, ambient occlusion, displacement, specular, and opacity. These maps are connected to the corresponding input slots of the Principled BSDF shader through Blender's node system. After applying the textures to all objects within a furniture group, a texture-variant image $I_t$ is rendered, after which the original textures are restored. Texture-related metadata, including texture name, category and tags are recorded for each variant. For each selected furniture group, we generate $K = 3$ global texture replacement variants.

Each texture edit is rendered as an independent operation. After rendering, the original texture parameters are restored to avoid interference between different samples.

### B.3. Vision-Guided Instruction Generation

To align texture edits with natural language instructions, we adopt a vision-guided multi-step refinement strategy (Wei et al., 2025), implemented using the Qwen3-VL model (Bai et al., 2025) and the SAM3 model (Carion et al., 2025).

For each image pair $(I, I_e)$, we first use Qwen3-VL to generate an initial instruction $P_0$ from metadata. Since metadata-only instructions may be incomplete or hallucinated, we apply a vision-guided refinement process. We compute the difference map $D = |I_e - I|$, binarize it to obtain a change mask $M_d$. The edited image $I_e$ and $M_d$ are then fed into SAM3 to obtain a precise object mask $M_s$, based on which a highlighted image $I_h$ is generated. Finally, $(I_h, I_e)$ is used to extract an object description $d$, and $(I, P_0, d)$ are jointly input to Qwen3-VL to produce the final concise instruction $P$.

Input Image      Nano Banana Pro      **TexEditor**

*Replace the dog's fur with a stylized rock tile texture, preserving its shape and posture, using a rough, layered stone pattern with subtle color variations to mimic natural rock formations.*

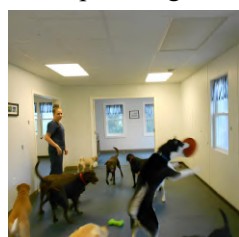 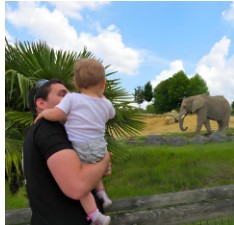 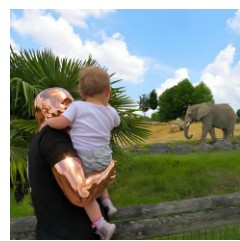 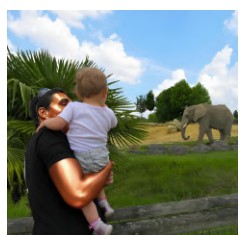

*Make the person's skin smooth and reflective with a copper texture, focusing on the face and arms, while keeping the shape unchanged.*

*Figure 11.* Failure cases

### B.4. Quality Filtering

The final stage applies a unified quality filtering process to all generated samples to remove unqualified data. The filtering criteria consist of three components. We use Qwen3-VL to assess the appearance difference between the image pair $(I, I_e)$ and obtain a score $s$. If $s$ falls below a threshold $\theta$, indicating that the texture change is either too subtle or visually implausible, the sample is discarded. Only samples that satisfy the criteria are retained and added to the final dataset, together with their refined instruction $P$.

### B.5. Output Dataset

After completing the full pipeline, the resulting dataset consists of structurally consistent, visually realistic, and semantically aligned indoor scene image pairs. Each sample contains a pre-edit and post-edit image pair $(I, I_e)$, together with the corresponding natural language editing instruction $P$. The dataset can be directly used for supervised fine-tuning of indoor scene editing models.

### B.6. Algorithm Summary

The complete pipeline is formalized in Algorithm 1 below.

## C. Failure case analysis

Although our method, TexEditor, can well preserve the structural consistency of the original subject during texture editing, it often misses certain targets when the instructions require editing multiple objects, as shown in Figure 11. Neither our method nor Nano Banana Pro can edit all the specified dogs and humans in the scene. Multi-object texture editing thus remains a meaningful and challenging direction for future research.

## D. TexBench Data and RL Training data Construction

TexBench is constructed using a semi-automated data generation pipeline that integrates image collection, instruction synthesis, and quality control to efficiently produce reliable image–instruction pairs, as illustrated in Figure 12. We first collect images and corresponding object masks from the COCO dataset (Lin et al., 2014), randomly select one annotated object per image, and generate texture editing instructions using a predefined texture editing vocabulary combined with the Qwen3-VL model.

To avoid distortion and information loss caused by directly resizing images with varying aspect ratios, we adopt an image

---

**Algorithm 1** Texture Editing Data Generation Pipeline

---

1: **Input:** 3D-FRONT room set $\mathcal{R}$, PBR texture library $\mathcal{T}$ (MatSynth)
2: **Output:** Image pairs $(I, I_e)$ with editing instructions $P$
3: **Parameters:** Max furniture attempts $N_{\text{furniture}}$, max camera attempts $N_{\text{camera}}$, texture variants per group $K$
4: Load the Qwen3-VL model and SAM3 model
5: **for** each room $r \in \mathcal{R}$ **do**
6:     Load 3D-FRONT scene of room $r$
7:     Remove objects with spatial conflicts via collision detection
8:     Collect editable furniture objects $\mathcal{O}$
9:     **for** grouping strategy $s \in \{\text{name+texture, name only}\}$ **do**
10:         Group $\mathcal{O}$ by strategy $s$ into furniture groups $\{\mathcal{G}_k\}$
11:         **for** furniture attempt $= 1$ to $N_{\text{furniture}}$ **do**
12:             Randomly select a target furniture group $\mathcal{G}$
13:             **for** camera attempt $= 1$ to $N_{\text{camera}}$ **do**
14:                 Sample camera pose around the spatial bounds of $\mathcal{G}$
15:                 **if** $\mathcal{G}$ clearly visible in camera view **then**
16:                     **break**
17:                 **end if**
18:             **end for**
19:             **if** camera sampling failed $N_{\text{camera}}$ times **then**
20:                 **continue** {Try next furniture}
21:             **end if**
22:             Fix camera pose for subsequent rendering
23:             Render original reference image $I$
24:             **for** each property $p \in \{\text{roughness, metalness, alpha}\}$ **do**
25:                 Compute average value $\bar{p}$ across all parts in $\mathcal{G}$
26:                 Consistently modify texture property $p$ of $\mathcal{G}$
27:                 Render property-variant image $I_p$
28:                 Restore original textures values
29:                 Record modification metadata (type, direction, value)
30:             **end for**
31:             **for** $i = 1$ to $K$ **do**
32:                 Sample random PBR texture $t$ from $\mathcal{T}$
33:                 Replace texture of $\mathcal{G}$ with PBR texture $t$
34:                 Render texture-variant image $I_t$
35:                 Restore original texture
36:                 Record texture metadata (name, category)
37:             **end for**
38:             Collect all image pairs $\mathcal{P} = \{(I, I_p)\} \cup \{(I, I_t)\}$
39:             **for** each image pair $(I, I_e) \in \mathcal{P}$ **do**
40:                 Extract metadata $M$ (edit type, parameters, texture info)
41:                 Input $(I, I_e, M)$ to Qwen3-VL
42:                 Generate initial editing instruction $P_0$
43:                 Compute image difference: $D = |I_e - I|$
44:                 Convert to grayscale and binarize to get change mask $M_d$
45:                 Input $I_e$ and $M_d$ to SAM3 for segmentation
46:                 Obtain precise segmentation mask $M_s$ of target object
47:                 Generate red-highlighted image $I_h$ on $I_e$ by $M_s$
48:                 Input $(I_h, I_e)$ to Qwen3-VL with identification prompt
49:                 Extract target object description $d$
50:                 Input $(I, P_0, d)$ to Qwen3-VL with refinement prompt
51:                 Generate refined concise instruction $P$ (word-limited)
52:                 Evaluate appearance difference using Qwen3-VL: $(I, I_e) \rightarrow s$
53:                 **if** $s < \theta$ (texture change too minimal or appearance implausible) **then**
54:                     Discard sample
55:                 **else**
56:                     Accept sample with instruction $T$
57:                 **end if**
58:             **end for**
59:             **break**
60:         **end for**
61:     **end for**
62: **end for**

---

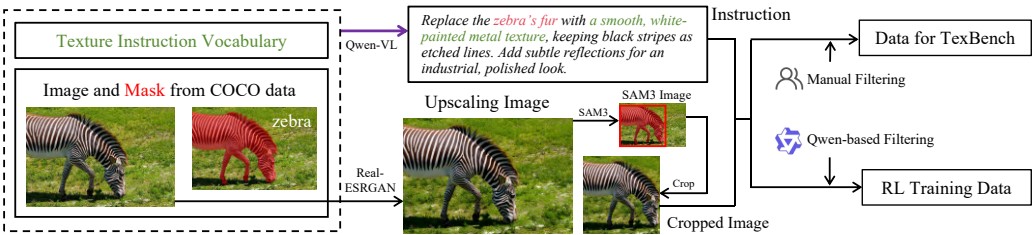

*Figure 12.* Pipeline for TexBench Data and RL Image–Instruction Pair Construction

cropping strategy. Specifically, image quality is first enhanced using Real-ESRGAN (Wang et al., 2021). We then apply SAM3 segmentation to extract the maximal bounding rectangle of the target object and crop the image accordingly.

A subset of the data is manually curated to obtain high-quality image–instruction pairs, which form the TexBench collection. For the remaining data, we apply an automatic filtering process based on Qwen to remove samples with inconsistent or unreliable image–instruction alignment, resulting in a large-scale image–instruction dataset for reinforcement learning.

# E. Human Evaluation Protocol

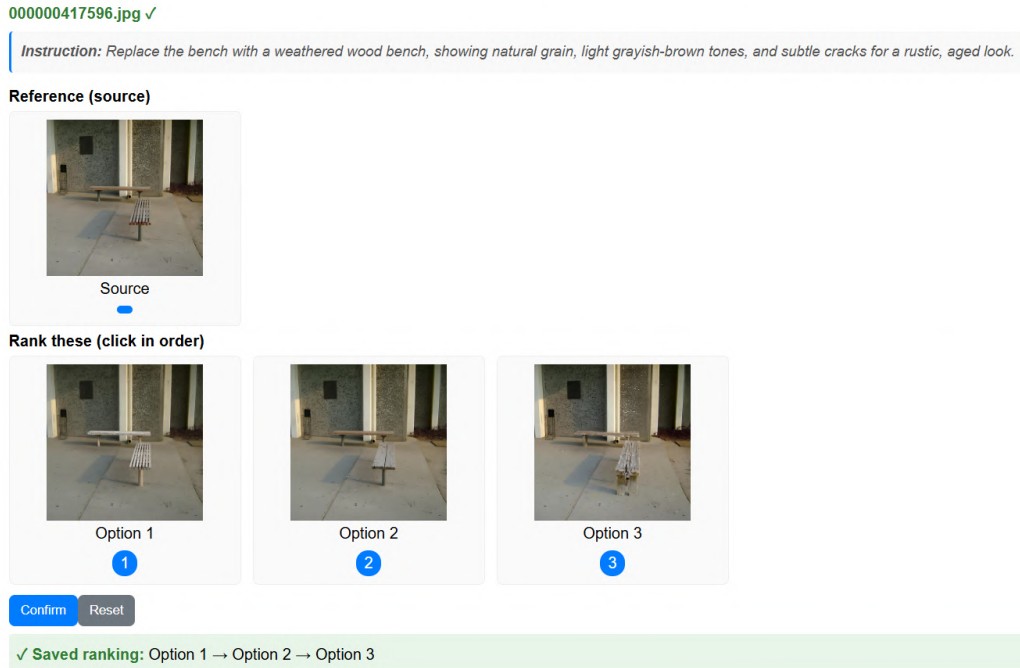

*Figure 13.* Screenshot of the web-based human evaluation interface

To validate the effectiveness of TexBench in evaluating texture editing performance, we design a Human Preference Ranking protocol to examine whether model rankings produced by TexBench are consistent with human subjective judgments. We evaluate seven models—Nano Banana Pro, Nano Banana (DeepMind, 2025b), Qwen-2509, Qwen-2511 (Wu et al., 2025), TexEditor, TexEditor-Base, and Alchemist (Sharma et al., 2024)—by generating texture editing results under identical input images and editing instructions.

We develop a web-based evaluation interface in which evaluators are asked to rank the results of three randomly selected models from best to worst according to texture realism, visual quality, and consistency with the editing instruction, without access to model identities, as shown in Figure 13. Five evaluators participate independently, and each human ranking is compared with the TexBench ranking for the same model subset.

We adopt ranking consistency accuracy as the evaluation metric, defined as the proportion of cases in which the TexBench ranking matches the human ranking. This evaluation measures the alignment between TexBench and human texture editing preferences, and assesses the reliability of TexBench as an automated benchmark.

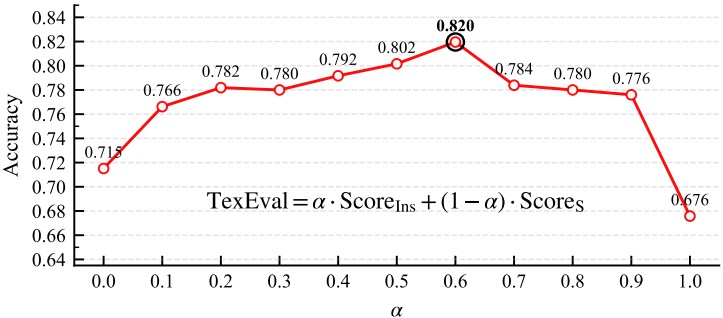

*Figure 14.* TexEval Performance under Different $\alpha$ Settings

TexEval is defined as a weighted combination of two components:

$$\text{TexEval} = \alpha \cdot \text{Score}_{\text{Ins}} + (1 - \alpha) \cdot \text{Score}_{\text{S}},$$

where $\alpha$ controls the relative importance between instruction adherence and structure preservation. We conduct a sensitivity analysis over $\alpha$ and compute the ranking consistency accuracy between TexEval-$\alpha$ and the human preference rankings described above, as shown in Figure 14.

We observe that the variant with $\alpha = 0.6$ achieves the highest consistency with human judgments. In contrast, when $\alpha = 0.0$ (i.e., relying solely on the instruction-related score) or $\alpha = 1.0$ (i.e., relying solely on the structure-related score), the accuracy is notably lower. These results indicate that effective texture editing evaluation requires balancing instruction compliance and structural fidelity, and that overemphasizing either aspect leads to reduced alignment with human perception. Based on this analysis, we adopt TexEval-0.6 as the default evaluation metric.

