# OpenReview forum: "TexEditor: Structure-Preserving Text-Driven Texture Editing"
_ICML.cc/2026/Conference — ICML 2026 regular_

### Official Review · Reviewer_9XXq · 2026-03-11

**Soundness:** 3
**Presentation:** 3
**Significance:** 2
**Originality:** 2
**Overall Recommendation:** 4
**Confidence:** 4

**Summary:**

This paper studies text-driven texture editing with an emphasis on preserving object geometry and scene structure while changing only appearance-related properties. The authors build TexEditor on top of Qwen-Image-Edit-2509 using a two-stage pipeline: supervised fine-tuning on a synthetic Blender-based dataset, TexBlender, where geometry is fixed and only texture is changed, followed by reinforcement learning on real-image queries with a reward that combines instruction following and structural consistency. The paper also introduces TexBench, a COCO-based benchmark for real-world texture editing with both texture replacement and attribute editing tasks, and TexEval, a metric that combines instruction adherence with structural preservation. Experiments on TexBench, a Blender-based benchmark, and ImgEdit show that the proposed system improves over the base model and strong baselines especially on structure-preserving texture edits.

**Compliance With Llm Reviewing Policy:**

Affirmed.

**Final Justification:**

The authors addressed my concern, so I remain positive about the paper and keep my score at 4.

**Key Questions For Authors:**

- The paper defines the structural reward in a somewhat abstract form and then instantiates it with SAUGE plus SSIM and task-specific normalization. Could the authors clarify the exact reward used during RL, including sign conventions, normalization, and whether the same structure signal is used in both training and TexEval?
- How independent is TexBench from the RL training data in practice, given that the RL data is constructed using the same general methodology? A strong clarification on overlap, distributional similarity, and why this does not bias the reported gains would increase my confidence in the external validity of the evaluation.

**Limitations:**

Yes.

**Strengths And Weaknesses:**

Strengths

- The overall method is technically sensible. The paper uses clean synthetic supervision in TexBlender to teach structure-preserving texture changes, then uses RL on real images to transfer this behavior with a reward that jointly encourages instruction following and structural consistency.
- The experimental evidence is fairly comprehensive. The paper reports results on both a new real-image benchmark and a Blender-based benchmark, includes qualitative results, runs ablations over SFT data and RL loss design, and also checks generalization on ImgEdit.
- The paper contributes a training pipeline, a synthetic dataset, a real-world benchmark, and an evaluation metric, and these components fit together coherently around one well-motivated problem.

Weaknesses

- The paper tightly couples training and evaluation. The RL data is constructed with the same general pipeline as TexBench, and TexEval also reuses the same structural signal family used in training. This slightly reduces confidence in how independent the evaluation is.
- The paper is understandable overall, but a few implementation details would benefit from clearer explanation.

---

> ### Author Rebuttal · Authors · 2026-03-31
>
> ### A1: Clarification of RL Formulation and external validity
>
> We note that the structural signal used during RL training is same as TexEval. To provide a more comprehensive and independent evaluation, we further introduce an external validation via a user study, where multiple models are assessed by human annotators.
>
> Specifically, we conducted a user study with three human annotators on 50 examples, including 25 texture editing cases and 25 attribute editing cases. For each case, annotators were shown the source image, the editing instruction, and the edited result, and evaluated the output from three perspectives: instruction following, structural preservation, and overall subjective quality. Instruction following was rated on a 1--5 scale from Not Applied to Perfectly Applied, while structural preservation was rated on a separate 1--5 scale from Completely Changed to Fully Preserved. Annotators also provided an overall subjective score reflecting their holistic judgment of the editing quality. We report the average scores across the three annotators over all evaluation examples.
>
> | Method           | Instruction | Structure | Overall Score |
> |------------------|-------------|-----------|---------------|
> | Qwen-2509        | 3.86        | 3.06      | 2.94          |
> | ALchemist        | 3.88        | 3.02      | 2.90          |
> | TexEditor-Base   | 3.92        | 3.37      | 3.20          |
> | Qwen-2511        | 4.14        | 3.58      | 3.50          |
> | Nano Banana      | 3.78        | 3.32      | 3.04          |
> | Nano Banana Pro  | **4.24**        | 3.64      | 3.68          |
> | **TexEditor**    | 4.10   | **4.26**  | **3.88**      |
>
>
> ### A2: Independence
>
> To examine data independence, we analyzed the overlap between TexBench and the RL training set. Although there is limited image-level overlap (3% in Texture testset 2% in another), we further applied a strict two-stage filtering protocol—coarse screening with Gemini 3-Flash followed by manual verification requiring both the editing subject and editing direction to match. Under this criterion, the exact image+instruction overlap is only 1 case (0.2%) in Texture and 0 cases (0%) in Attribute. This suggests that semantic overlap is negligible.
>
> Although both training and evaluation are grounded in COCO, the combinatorial variation in image content and editing directions makes simple memorization unlikely. This is also reflected in the negligible exact image+instruction overlap reported above. Therefore, the risk of overfitting to specific instances is limited.

---

> > ### Author Rebuttal · Reviewer_9XXq · 2026-04-01
> >
> > The authors addressed my concern, so I remain positive about the paper and keep my score at 4.

---

> > > ### Author Response · Authors · 2026-04-03
> > >
> > > Thank you for reading our rebuttal. We would like to clarify that the main novelty of this work is not a single technical module, but the identification of a key limitation of current image editing models: preserving structure during texture editing. Based on this observation, we develop a more systematic solution around this problem, including the method, benchmark, and evaluation framework. By making this limitation more explicit and measurable, we hope the paper can also support future progress on this capability through clearer evaluation and comparison. We expect this clarification make the paper’s novelty, significance, and potential value to the community more clearly reflected in the final assessment.

---

### Official Review · Reviewer_s7WX · 2026-03-13

**Soundness:** 3
**Presentation:** 3
**Significance:** 2
**Originality:** 2
**Overall Recommendation:** 4
**Confidence:** 3

**Summary:**

This paper addresses the problem of text-driven texture editing while preserving the geometric structure of objects. The authors observe that existing image editing models, including commercial systems like Nano Banana Pro, frequently distort object geometry when performing purely appearance-related edits. To tackle this, they propose TexEditor, built on Qwen-Image-Edit-2509, with a two-stage training strategy: supervised fine-tuning on a synthetic dataset (TexBlender) followed by reinforcement learning with a structure- preserving loss (StructureNFT).
TexBlender is constructed using Blender and 3D-Front assets, generating paired images where only textures differ while geometry remains identical. The RL stage uses a hybrid reward combining MLLM-based instruction-following scores and wireframe-based structural consistency (SAUGE + SSIM). The paper also introduces TexBench, a COCO-based real-world benchmark with 825 human-verified queries, and TexEval, a composite metric combining instruction adherence with structural consistency scores.

**Compliance With Llm Reviewing Policy:**

Affirmed.

**Key Questions For Authors:**

1. How sensitive is the method to the normalization thresholds (τ_min, τ_max)? If applied to a new texture editing domain (e.g., outdoor scenes, non-rigid objects), would these need to be re-tuned?

2. Can you provide a more detailed analysis separating the effect of RL data quantity vs. the StructureNFT design? For instance, what happens with 5× more RL data but without the structure loss?

**Limitations:**

The authors discuss limitations briefly in the impact statement, mentioning potential misuse for manipulating visual content. However, they do not discuss failure cases of the method itself (e.g., when structure preservation conflicts with the editing intent) or computational overhead of the RL training. A more thorough discussion of method-specific limitations would be beneficial.

**Strengths And Weaknesses:**

Strengths:
1. The two-stage training pipeline (SFT → RL) is well-motivated: synthetic data provides clean structure-preserving supervision, and RL bridges the sim-to-real gap. The ablation study (Table 4, Configs A–I) systematically validates each component.
2. The paper is clearly written with a logical flow. Figure 1 effectively motivates the problem, and the pipeline figures (Fig. 2, 3) are informative.
3. The TexBench benchmark and TexEval metric fill a genuine gap— prior texture editing benchmarks (e.g., Alchemist) are limited to synthetic single-object scenes and lack structure-aware evaluation.
4. Combining wireframe-based structural loss (SAUGE + SSIM) with MLLM reward in RL for texture editing is a novel formulation. The insight that existing MLLMs cannot detect subtle structural changes (Fig. 5) is valuable.

Weaknesses:
 1. The normalization thresholds in Eq. 6–7 (τ_min, τ_max) are determined empirically and differ per subtask. The authors acknowledge optimality is not guaranteed. This introduces fragile hyperparameters that may not transfer to new domains.
2. The RL training uses only 1,500 COCO images. The generalization study (Sec. 4.4) shows performance drops on non-texture tasks (Background, Style), likely due to limited RL data coverage rather than fundamental method limitations—but this is not well-disentangled.
3. The task scope is narrow: only texture/material editing on objects. The method does not address broader editing capabilities (shape, layout, color). While the paper claims generalization on ImgEdit, gains are concentrated on texture-related subtasks.
4. The core architecture is unchanged from Qwen-Image-Edit-2509. The novelty lies primarily in the training data construction and RL reward design, which may be seen as engineering contributions rather than fundamental methodological advances.
5. TexEval (Eq. 8) is a simple linear combination (α-weighted) of instruction score and structure score. The user study validates it outperforms individual metrics, but the paper does not explore non-linear alternatives or analyze sensitivity to α.

---

> ### Author Rebuttal · Authors · 2026-03-31
>
> ### A1: Threshold sensitivity
>
> Regarding the reviewer’s concern about the sensitivity of the structural loss hyperparameters, we would like to clarify three points.
>
> First, the structure score computed by SSIM naturally follows a specific data distribution. Based on our analysis of Qwen-Edit outputs, around 80% of samples fall within the range of 0.65–0.90. Therefore, aligning the hyperparameter design with this distribution (i.e., applying a smoother thresholding strategy) is necessary for stable optimization.
>
> Second, we addtionally evaluate the sensitivity of these thresholds. In the original setting, for the attribute task, $\tau_{\min}$ and $\tau_{\max}$ are 0.8 and 0.95, and for the texture task, they are 0.7 and 0.9. In the new setting, we relax the lower bound by decreasing $\tau_{\min}$ by 0.1 for both tasks, while keeping $\tau_{\max}$ unchanged. For a controlled comparison, we fix the RL training budget across all settings: 40 update steps, with 24 prompts per step and 6 rollouts per prompt.
>
> As shown in the results, Base-40 corresponds to the original hyperparameter setting, while Hyper-40 uses the relaxed thresholds. Lowering $\tau_{\min}$ effectively relaxes the structural constraint, leading the model to place more emphasis on instruction following. However, the overall performance remains largely unchanged, indicating that the method is relatively robust to this hyperparameter choice.
>
> Finally, Our TexBench benchmark is constructed from COCO images, which cover diverse real-world scenes and object categories. Therefore, the training and evaluation data are already grounded in a broad distribution, reducing the risk of domain overfitting. As a result, we do not expect significant generalization issues when transferring to other natural image settings, except for highly specialized or narrowly defined sub-domains.
>
> ### A2: Data quantity vs structure loss
> Regarding the relationship between RL training data and the structural loss, we conduct additional analysis under the same experimental setup as in A1. Specifically, we scale up the RL training data and extend training without the structural loss, ensuring that the model is always trained on new data (i.e., not completing a full epoch).
>
> As shown in the table, No-Struct-40, No-Struct-60, and No-Struct-80 correspond to 40, 60, and 80 update steps under this setting. We observe that simply increasing data and computation—even doubling both—still fails to match the performance of Base-40, with a more pronounced gap on attribute editing tasks that require stronger structural preservation. Moreover, the performance gains begin to saturate as training progresses.
>
> These results suggest that scaling data alone provides limited benefit, and the structural loss plays a critical role in achieving strong performance.
>
>
>
> | Config | Attribute Ins | Attribute Structure | Attribute TexEval | Texture Ins | Texture Structure | Texture TexEval |
> |---|---:|---:|---:|---:|---:|---:|
> | Base-40 | 0.625 | 0.864 | 0.733 | 0.821 | 0.898 | 0.856 |
> | No-Struct-40 | 0.704 | 0.363 | 0.551 | 0.802 | 0.728 | 0.769 |
> | No-Struct-60 | 0.738 | 0.449 | 0.608 | 0.814 | 0.761 | 0.790 |
> | No-Struct-80 | 0.741 | 0.492 | 0.629 | 0.820 | 0.782 | 0.803 |
> | hyper-40 | 0.690 | 0.795 | 0.737 | 0.825 | 0.868 | 0.844 |
>
>
> ### A3: Limitations
> We have pointed some limitations of our method in the main text and will emphasize them clearly in the revised manuscript. As discussed in Appendix A.3, we evaluate our model on the general editing benchmark ImgEdit and observe weaker performance on sub-tasks that require significant structural changes to the main object. In addition, in Appendix C, we provide a failure case analysis, showing that our method (as well as existing approaches) struggles with multi-object texture editing scenarios.
>
> ### A4: $\alpha$ weighting
> We analyze the impact of the weighting factor α in Appendix E. As shown in Fig. 14, the performance remains stable across a range of α values, indicating that TexEval is not sensitive to the exact choice of α. In addition, the user study further validates that the selected α achieves better alignment with human judgments than individual metrics.
>
> ### A5: Practical cost
> To clarify practical cost, we will report a concise compute summary in the revised paper. TexEditor is trained with LoRA-based SFT on 10,000 synthetic pairs for 2 epochs, followed by RL on 1,500 COCO-based queries for 1 epoch and 6 rollouts per prompt.
>
> ### Clarification on contribution scope:
> We agree that our work does not introduce a new backbone. Our contribution instead lies in a problem-specific framework for structure-preserving texture editing, spanning synthetic supervision, structure-aware RL, and a dedicated benchmark and metric. Importantly, we believe this design—focusing on training strategy and evaluation rather than model architecture—is more easily adoptable by existing methods, and thus more conducive to reuse and broader impact within the community.

---

### Official Review · Reviewer_1cau · 2026-03-13

**Soundness:** 3
**Presentation:** 3
**Significance:** 3
**Originality:** 2
**Overall Recommendation:** 4
**Confidence:** 4

**Summary:**

This paper proposes a new text-driven image texture (appearance) editing method. To support training, the paper presents a new synthetic dataset, TexBlender, that uses Blender to render images based on 3D-FRONT with material manipulation. The main pipeline first involves supervised fine-tuning using the synthetic dataset, followed by reinforcement learning with rewards combining multiple factors. The paper makes extensive contributions to different stages of texture editing research, including a new synthetic dataset TexBlender that uses blender and 3D-FRONT data for generating results with material editing; a Reinforcement Learning based pipeline for texture editing TexEditor, and a benchmark dataset TexBench, a general-purpose real-world benchmark for text-guided texture editing. For TexBench, it further introduces TexEval, a composite metric that combines low-level visual structural cues with MLLM instruction adherence, providing a more holistic assessment of texture editing quality.

**Compliance With Llm Reviewing Policy:**

Affirmed.

**Final Justification:**

The rebuttal addressed my concerns, so I remain positive with the paper.

**Key Questions For Authors:**

- How/Why the model trained on TexBlender is generalizable to other editing tasks?

**Limitations:**

The paper discusses multi-object editing as a limitation which is fine. How about other challenging cases, such as sophisticated material properties, complicated lighting/reflections, etc.?

**Strengths And Weaknesses:**

Strengths:
- The paper makes extensive contributions to different stages of texture editing research, including a new synthetic dataset TexBlender that uses blender and 3D-FRONT data for generating results with material editing; a Reinforcement Learning based pipeline for texture editing TexEditor, and a benchmark dataset TexBench, a general-purpose real-world benchmark for text-guided texture editing. For TexBench, it further introduces TexEval, a composite metric that combines low-level visual structural cues with MLLM instruction adherence, providing a more holistic assessment of texture editing quality.
- The proposed method is generally plausible, and achieves competitive results.
- The dataset and benchmark can make clear contributions to the community, if they will be publicly released. I think it is not clearly promised at this stage, but from the context, the paper seems to imply the work will be open.

Weaknesses:
- Most parts of the technical contributions are fairly straightforward, so it is more like engineering effort rather than novel, original research.
- The TexBlender dataset is only based on indoor scenes (3D-FRONT). It is unclear how/why the dataset and models trained on it are sufficiently generalizable. Moreover, it is worth discussing why such generalization is feasible.
- Texture may have different meanings, and in most cases, the proposed method only edits appearance, rather than actually textures (although this is a term used in certain context).

---

> ### Author Rebuttal · Authors · 2026-03-31
>
> We thank the reviewer for the positive assessment of our work. We commit to open-sourcing our work upon acceptance, including **TexBench**, **TexEditor**, the **scripts for constructing the Blender-based dataset**, as well as other components that may be useful to the community. This release plan has already received internal approval.
>
>
> ### A1: Why can TexBlender generalize beyond indoor synthetic scenes?
> TexBlender is not intended to provide full domain coverage. Instead, its role is to offer clean supervision for learning a transferable editing prior—specifically, modifying appearance while preserving geometry, layout, and object identity. This prior is largely domain-agnostic and can generalize beyond indoor scenes.
>
> This design is reflected in the ablation results (Table 4). Transitioning from Config A (Qwen-Edit backbone only) to Config C (further trained on TexBlender) leads to noticeable improvements in texture editing performance, indicating that the learned prior is indeed effective. However, generalization to more challenging real-world scenarios, as evaluated on TexBench, remains limited at this stage. By further introducing reinforcement learning on real-world data constructed from COCO—guided by both structural consistency and instruction-following rewards—we substantially strengthen this capability, transforming it from occasional success into consistently reliable performance.
>
> ### A2: Other challenging cases
> We agree that sophisticated material properties and complex lighting or reflection effects are important challenging cases. Based on our current empirical observations, however, they do not appear to be the dominant failure mode in our present benchmark.  As shown in Fig of the main paper. 6, cases involving specular highlights (row 3), diffuse shading (row 4), and reflections (row 6) are generally handled well. This observation holds not only for our method but also for several baseline models.
>
> In contrast, more frequent failures are related to structural drift or incomplete target coverage, especially in multi-object editing cases. We will clarify this point in the revision and explicitly discuss that more extreme material and illumination cases remain an important direction for future benchmark expansion.
>
> ### A3: Clarification on novelty and terminology
> We agree that some individual components build on existing tools. Our main contribution is therefore not a single isolated algorithmic novelty, but a unified treatment of a previously under-addressed problem: structure-preserving texture editing. The paper contributes a coherent solution spanning data construction, training, and evaluation, together with a benchmark and metric designed for this specific failure mode. We will revise the paper to make this positioning clearer. We will also clarify our use of the term “texture”: in this work, we use it in the broader image-editing sense to cover appearance-related surface changes, including both material replacement and attribute changes such as roughness, metallicity, and transparency, to avoid ambiguity with narrower graphics-specific usage.

---

> > ### Author Rebuttal · Reviewer_1cau · 2026-04-03
> >
> > The authors addressed my concern, so I remain positive about the paper and keep my positive score.

---

### Official Review · Reviewer_UCFt · 2026-03-14

**Soundness:** 3
**Presentation:** 3
**Significance:** 3
**Originality:** 3
**Overall Recommendation:** 4
**Confidence:** 4

**Summary:**

This paper studies text-guided texture editing, with the emphasis on changing appearance while preserving the underlying object structure. The authors argue that current editing systems often regenerate or deform the target object even when the requested change is purely texture-related. To address this, they propose TexEditor, built on Qwen-Image-Edit-2509, together with two main ingredients: a Blender-generated SFT dataset, TexBlender, designed to provide clean structure-preserving supervision, and a reinforcement-learning stage, StructureNFT, which transfers this behavior to real images through instruction-following and structure-preservation rewards. The paper also introduces TexBench and TexEval for evaluating real-world texture editing, and reports results on both its new benchmark and existing datasets, with an additional generalization study on ImgEdit.

**Compliance With Llm Reviewing Policy:**

Affirmed.

**Key Questions For Authors:**

1. How sensitive are the RL results to the exact reward design, especially the balance between instruction-following and structure-preservation terms, and to the choice of Gemini as the instruction reward model?
2. Since TexBench and TexEval are both introduced in this paper, can the authors provide stronger external validation, for example through human preference comparisons against the main model ranking or broader evaluation on pre-existing real-image editing benchmarks?
3. The ImgEdit results suggest weaker transfer on categories such as Action and Add. What kinds of edits are most likely to break the structure-preserving behavior learned by TexEditor?

**Limitations:**

The paper discusses the motivation and empirical gains well, but it should be more explicit about reward-model dependence, benchmark self-containment, and the cost of the full training pipeline.

**Strengths And Weaknesses:**

**Strengths**
- The problem is well chosen. Texture editing should be easier than full object regeneration in principle, yet existing systems often distort shape or boundaries; the paper isolates this gap clearly.
- The method has a coherent design. Clean synthetic supervision for cold start, followed by RL on real images with an explicit structure term, is a sensible way to separate structure preservation from appearance manipulation.
- The benchmark contribution is meaningful. TexBench and TexEval make the submission more than a model paper, and the paper makes a reasonable case that existing texture-editing benchmarks are too narrow or too synthetic.
- The empirical results are strong. On TexBench, TexEditor improves over strong baselines including Nano Banana Pro on both texture and attribute editing, and the ablation table is useful in showing that both the SFT data design and the RL objective matter.

**Weaknesses**
- A large part of the empirical story depends on artifacts introduced by the authors themselves: TexBlender, TexBench, and TexEval. That is not inherently problematic, but it makes external validation more important than it currently is.
- The RL stage relies on a fairly composite reward design, including instruction-following rewards from Gemini and structure terms based on vision tools. The paper does not say enough about sensitivity to reward choice, reward weights, or reward-model bias.
- The ImgEdit generalization study is mixed rather than uniformly strong. The paper improves on texture-related categories such as Background, Adjust, and Compose, but is weaker on less related categories such as Action and Add.
- Practical cost is not very clear. Given the use of synthetic data generation, RL, and multiple external components, I would expect a sharper account of training complexity and the deployment overhead relative to the underlying editor.

---

> ### Author Rebuttal · Authors · 2026-03-31
>
> ### A1: Reward sensitivity
> We address this concern from two complementary angles. First, the main paper already includes ablations over different RL loss combinations and the regularized vs. unregularized structure term, showing that the full objective yields the best balance. Second, in response to the reviewer, we additionally test reward-weight perturbations and an alternative Gemini API, and observe that the overall conclusion remains stable.
> For a controlled comparison under computational budget: we fixed the RL training schedule to 40 update steps for all settings. In each update step, we sampled 24 distinct prompts and performed 6 rollouts.
>
> The first row corresponds to the default loss weighting used in our paper, where the instruction and structure rewards are combined with a ratio of 0.55:0.45. The second and third rows examine two shifts of this weighting, placing relatively more emphasis on structure or instruction, respectively (3:7 and 8:2). Notably, both settings deviate from the default weighting by the same margin of 0.25, making them symmetric around our original choice. The results show that changing the loss weighting does affect individual metric scores, but has only a limited impact on the overall score.
>
> In addition, we replaced the reward API from Gemini-3-Flash-Preview with Gemini-2.5-Flash. We observe only a slight performance drop under this change, which suggests that our method is reasonably robust to the choice of reward API.
>
> | Config       | Texture Ins | Texture Structure | Texture TexEval | Attribute Ins | Attribute Structure | Attribute TexEval |
> |--------------|-------------|-------------------|-----------------|---------------|---------------------|-------------------|
> | Base         | 0.821       | 0.898             | 0.856           | 0.625         | 0.864               | 0.733             |
> | Ins3:Strct7 | 0.748       | 0.964             | 0.845           | 0.541         | 0.943               | 0.722             |
> | Ins8:Strct2  | 0.804       | 0.827             | 0.814           | 0.714         | 0.736               | 0.724             |
> | Gemini2.5    | 0.803       | 0.882             | 0.839           | 0.623         | 0.851               | 0.726             |
>
>
> ### A2: External validation
>
> We agree external validation is important, since TexBench and TexEval are introduced in this paper. Beyond the human study already included in the paper for validating TexEval, we further conduct a new blind human evaluation directly on model outputs.
>
> We sample 50 real-image cases (25 texture and 25 attribute ). For each case, three annotators are shown the source image, the editing instruction, and anonymized outputs from different methods in randomized order, and rate each result on instruction following, structure preservation, and overall subjective quality (all on 1--5 scales). We report the average scores across the three annotators over all evaluation examples.The results show that TexEditor achieves the best structure preservation (4.26) and best overall score (3.88) among all compared methods, while remaining competitive in instruction following (4.10).
>
> | Method           | Instruction | Structure | Overall Score |
> |------------------|-------------|-----------|---------------|
> | Qwen-2509        | 3.86        | 3.06      | 2.94          |
> | ALchemist        | 3.88        | 3.02      | 2.90          |
> | TexEditor-Base   | 3.92        | 3.37      | 3.20          |
> | Qwen-2511        | 4.14        | 3.58      | 3.50          |
> | Nano Banana      | 3.78        | 3.32      | 3.04          |
> | Nano Banana Pro  | **4.24**        | 3.64      | 3.68          |
> | **TexEditor**    | 4.10   | **4.26**  | **3.88**      |
>
> ###A3: limitation
> A possible explanation is that TexEditor is not trained on large-scale general editing data, and tends to assume that the desired edit should preserve the original structure. This inductive bias is helpful for texture editing, but becomes a limitation for categories such as Action and Add, where the primary object often needs substantial structural change. We believe this limitation could be addressed in future work by extending the training data to more general editing scenarios.
>
> ### A4: Independence
> To examine data independence, we analyzed the overlap between TexBench and the RL training set. Although the image-level overlap is limited (3% in Texture testset 2% in another),  a strict two-stage filtering protocol—coarse screening with Gemini 3-Flash followed by manual verification is conducted. Under this criterion, the exact image+instruction overlap is only 1 case (0.2%) in Texture and 0 cases (0%) in Attribute. This suggests that semantic overlap is negligible.
> ### A5: Practical cost
> To clarify practical cost, we will report a concise compute summary in the revised paper. TexEditor is trained with LoRA-based SFT on 10,000 synthetic pairs for 2 epochs, followed by RL on 1,500 COCO-based queries for 1 epoch and 6 rollouts per prompt.

---

> > ### Author Rebuttal · Reviewer_UCFt · 2026-04-04
> >
> > The authors addressed my concern, so I remain positive about the paper and keep my score at 4.

---

### Decision · Program_Chairs · 2026-04-30

**Decision:**

Accept (regular)

**Comment:**

The overall recommendations are 4 weak accepts. The reviewers agreed that (1) the proposed pipleline is well-motivated and reasonable, (2) the dataset and benchmark make clear contributions to the community, , and (3) empirical results are strong. They raised some concerns about reward sensitivity, external validation, generalization, and data independency. The authors' rebuttal has fully resolved their concerns.